# Evolutionary transitions between beneficial and phytopathogenic *Rhodococcus* challenge disease management

Elizabeth A Savory[1†], Skylar L Fuller[1,2†], Alexandra J Weisberg[1†], William J Thomas[1], Michael I Gordon[1], Danielle M Stevens[1], Allison L Creason[1,2‡], Michael S Belcher[1], Maryna Serdani[1], Michele S Wiseman[1], Niklaus J Grünwald[3], Melodie L Putnam[1], Jeff H Chang[1,2,4]*

[1]Department of Botany and Plant Pathology, Oregon State University, Corvallis, United States; [2]Molecular and Cellular Biology Program, Oregon State University, Corvallis, United States; [3]Horticultural Crops Research Laboratory, United States Department of Agriculture and Agricultural Research Service, Corvallis, United States; [4]Center for Genome Research, Oregon State University, Corvallis, United States

*For correspondence:
changj@science.oregonstate.edu

†These authors contributed equally to this work

Present address:
‡Computational Biology, Oregon Health and Sciences University, Portland, United States

Competing interests: The authors declare that no competing interests exist.

**Abstract** Understanding how bacteria affect plant health is crucial for developing sustainable crop production systems. We coupled ecological sampling and genome sequencing to characterize the population genetic history of *Rhodococcus* and the distribution patterns of virulence plasmids in isolates from nurseries. Analysis of chromosome sequences shows that plants host multiple lineages of *Rhodococcus*, and suggested that these bacteria are transmitted due to independent introductions, reservoir populations, and point source outbreaks. We demonstrate that isolates lacking virulence genes promote beneficial plant growth, and that the acquisition of a virulence plasmid is sufficient to transition beneficial symbionts to phytopathogens. This evolutionary transition, along with the distribution patterns of plasmids, reveals the impact of horizontal gene transfer in rapidly generating new pathogenic lineages and provides an alternative explanation for pathogen transmission patterns. Results also uncovered a misdiagnosed epidemic that implicated beneficial *Rhodococcus* bacteria as pathogens of pistachio. The misdiagnosis perpetuated the unnecessary removal of trees and exacerbated economic losses.
DOI: https://doi.org/10.7554/eLife.30925.001

## Introduction

Symbioses are persistent and intimate interactions between organisms. In pathogenic interactions, one partner benefits at the expense of the other. In mutualistic symbioses, specific partners interact and reciprocally benefit. Associative symbioses are a variation of mutualism in which there is lower specificity between interacting partners (*Drogue et al., 2012*). In agricultural systems, practices are employed to limit pathogens, to introduce nitrogen-fixing mutualistic rhizobia and to restore associative symbionts such as plant growth-promoting bacteria (PGPB). PGPB can directly promote the growth of plants and protect against pathogens (*Barea et al., 2005*; *Pieterse et al., 2014*).

The beneficial or parasitic outcomes of symbioses, especially those involving environmentally acquired partners, are often not guaranteed. The health of the host, location of the symbiont on the host, or unregulated proliferation of the symbiont can lead to alternative outcomes (*Lin and Koskella, 2015*). The genotype of the symbiont is also a critical factor, as horizontal gene

**eLife digest** All organisms live in a world teeming with bacteria. Some bacteria are beneficial and, for example, provide their hosts with nutrients. Others cause harm, for example, by stealing nutrients and causing disease. Many bacteria can also gain DNA from other bacteria, and the genes encoded within the new DNA can help them to live with other organisms. This can start the bacteria on an evolutionary path to becoming beneficial or harmful.

*Rhodococcus* are bacteria that live in association with many species of plants, including trees. Most are harmless but some cause disease. Plants infected with harmful *Rhodococcus* can show deformed growth, which causes major losses to the nursery industry.

Savory, Fuller, Weisberg et al. set out to understand how disease-causing *Rhodococcus* are introduced into nurseries, if they are transferred between nurseries, whether they persist in nurseries, and how to limit their spread. It turns out that harmless *Rhodococcus* are beneficial to plants. However, if these harmless bacteria gain a certain DNA molecule – called a virulence plasmid – they can convert into harmful bacteria. Further analysis showed that some nurseries repeatedly acquired the harmful bacteria. The pattern of affected nurseries suggested that some might have purchased diseased plants from a common provider. In other cases, the sources remained a mystery.

Savory et al. also report that, contrary to previous findings, there is no evidence to support the diagnosis that *Rhodococcus* without a virulence plasmid are responsible for an unusual growth problem that has plagued the pistachio industry. In recent years, this incorrect diagnosis led to trees being unnecessarily destroyed, worsening the economic losses.

These findings suggest that genes moving between bacteria can dramatically change how those bacteria interact with the organisms in which they live. It needs to be shown whether this is an exceptional process, unique to only certain groups of bacteria, or if it is more widespread in nature. These findings could inform future disease management strategies to better protect agricultural systems.

DOI: https://doi.org/10.7554/eLife.30925.002

transfer (HGT) can lead to the acquisition of new genes that innovate genomes, driving evolutionary transitions and establishing new lineages of beneficial or pathogenic symbionts (*Soucy et al., 2015*). In some pathogenic symbionts, however, HGT does not bestow the genome with innovative functions, nor do these genomes exhibit substantive changes. Rather, the few horizontally acquired genes encode products that reprogram core genes, thereby co-opting the genome for virulence (*Letek et al., 2010*).

*Rhodococcus* is a genus of Gram-positive bacteria with members that persist in a variety of terrestrial and aquatic ecosystems (*de Carvalho et al., 2014*; *Larkin et al., 2005*). *Rhodococcus* includes taxonomic groups with members that have been repeatedly recovered from leaf and root tissues of various species of plants (*Bai et al., 2015*; *Bodenhausen et al., 2013*; *Bulgarelli et al., 2012*; *Hong et al., 2015, 2016*; *Lebeis et al., 2015*; *Lundberg et al., 2012*; *Qin et al., 2009, 2011*; *Salam et al., 2017*). It has been suggested that hosts enrich for members of *Rhodococcus* because of the beneficial traits of the bacteria (*Hong et al., 2015, 2016*).

Plant-associated *Rhodococcus* species are better known as pathogens (*Putnam and Miller, 2007*). Two clades of *Rhodococcus* include members that can cause disease in over 100 genera of plants (*Creason et al., 2014a2014a*; *Putnam and Miller, 2007*). Herbaceous plants are the most commonly affected whereas woody plants are less frequently infected. Disease symptoms include leafy galls, witches'-brooms, and other disfiguring growths. Pathogenic isolates of *Rhodococcus*, typified by the most-studied isolate D188, require three virulence loci that are most frequently found clustered on virulence plasmids (*Crespi et al., 1992*; *Stes et al., 2011*). These plasmids are approximately 200 kb in length and are linear replicons (*Francis et al., 2012*; *Creason et al., 2014b*). Some of the *fas* (*fasciation*) genes are necessary for disease and encode proteins that synthesize and modify cytokinins, which are predicted to be secreted effectors (*Crespi et al., 1994*; *Pertry et al., 2009*). The *fasR* gene, predicted to be a transcriptional regulator, is also necessary for pathogenicity (*Temmerman et al., 2000*). *att* (*attenuation*) mutants are reportedly attenuated in disease and thus implicated in virulence (*Crespi et al., 1992*; *Maes et al., 2001*). The *vicA* gene, which encodes

malate synthase, an enzyme in the glyoxylate cycle, is the only locus encoded on the chromosome implicated in virulence (*Vereecke et al., 2002*).

Pathogenic *Rhodococcus* are particularly problematic in agricultural settings that produce plants for their aesthetic value. A variety of biotic and abiotic stresses, some induced by anthropogenic practices, cause symptoms that are confused with those caused by *Rhodococcus* (*Putnam and Miller, 2007*). Furthermore, isolates of *Rhodococcus* that lack virulence genes are often cultured from symptomatic tissues (*Creason et al., 2014b*; *Nikolaeva et al., 2009*). Multiple tests are used to confirm that a plant is infected by pathogenic *Rhodococcus*. Bacteria must exhibit the proper morphology on selective media and be taxonomically assigned to *Rhodococcus*. The bacteria must have virulence genes and must cause disease symptoms in susceptible indicator plants. Because the *fasR* gene and some of the *fas* genes are necessary for pathogenicity of the bacteria, their detection is sufficient to confirm pathogenicity of *Rhodococcus* isolates.

In 2011, populations of micropropagated pistachio UCB-1 (*Pistacia atlantica* × *Pistacia integer-rima*) rootstocks planted in commercial fields began showing an unusual phenotype (*Stamler et al., 2015a*, *2015b*). Aerial phenotypes of 'pistachio bushy top syndrome' include shortened internodes, loss of apical dominance, stem galls, and reduced grafting success. Estimates suggest that more than 1 million trees grown on 25,000–30,000 acres were affected. *Rhodococcus* isolates were cultured from symptomatic plants, and when inoculated onto UCB-1, they caused morphological changes to the hosts (*Stamler et al., 2015b*). This was the first report of pistachio being susceptible to *Rhodococcus*. Subsequent release of the genome sequences for PBTS1 and PBTS2, the reported outbreak strains, indicated they lack virulence loci (*Stamler et al., 2016*). The detection of *vicA* was used as evidence for pathogenic *Rhodococcus* bacteria and to guide management practices, the most extreme and costly being the removal of entire orchards. A second incidence of pistachio bushy top syndrome occurred in 2016, resulting in the destruction of 1.5 million nursery trees.

We determined and analyzed genome sequences from over 80 isolates of *Rhodococcus*, mostly collected from symptomatic herbaceous plants grown in production settings. Analysis of chromosomal sequences shows that plants host multiple lineages of *Rhodococcus*. Isolates that lack virulence plasmids can promote changes to the architecture of roots, but if a virulence plasmid is acquired, the isolates transition to being pathogenic. The analysis of chromosomal sequences of pathogens revealed the potential for multiple infections and reservoir populations at nursery sites, as well as for point source outbreaks. However, the distribution patterns of virulence plasmids suggested that agricultural systems can be locations that promote evolutionary transitions and the rapid generation of new lineages of pathogens, providing an alternative route for the spread of pathogens. Last, our results challenge previous conclusions that *Rhodococcus* isolates lacking virulence genes are causative agents of pistachio bushy top syndrome, suggesting that the pistachio syndrome was likely misdiagnosed (*Stamler et al., 2015a*, *2015b*).

## Results

### Epidemiological links in nurseries

We used a genomic epidemiological approach to study the transmission patterns of *Rhodococcus* in a plant agricultural system. Sixty isolates were collected hierarchically across space and time. Multiple isolates were cultured from the same symptomatic tissues or from different plants grown at the same production site (*Supplementary file 1A*). Previously sequenced isolates, many also collected from production sites, were included (*Supplementary file 1A*; *Creason et al., 2014b*). The nursery sources of the isolates were anonymized.

Phylogenetic analysis placed the 60 isolates within *Rhodococcus* Clades I and II, which also included the 15 previously confirmed pathogenic isolates (*Figure 1*; *Creason et al., 2014b*). The two clades are sisters to Clades III and IV. Clade III consists of isolates that were cultured from microbiota of the model plant*Arabidopsis thaliana* (*Bai et al., 2015*; *Bodenhausen et al., 2013*; *Bulgarelli et al., 2012*; *Lebeis et al., 2015*; *Lundberg et al., 2012*). No member of Clades III or IV have virulence genes. The four clades are distinct, and separated by a long branch, from the other species of *Rhodococcus* (*Figure 1—figure supplement 1*). The isolates in the four clades were operationally classified into 17 species, indicated with a lowercase letter, of which 14 have at least one isolate cultured from a plant (*Supplementary file 1B*).

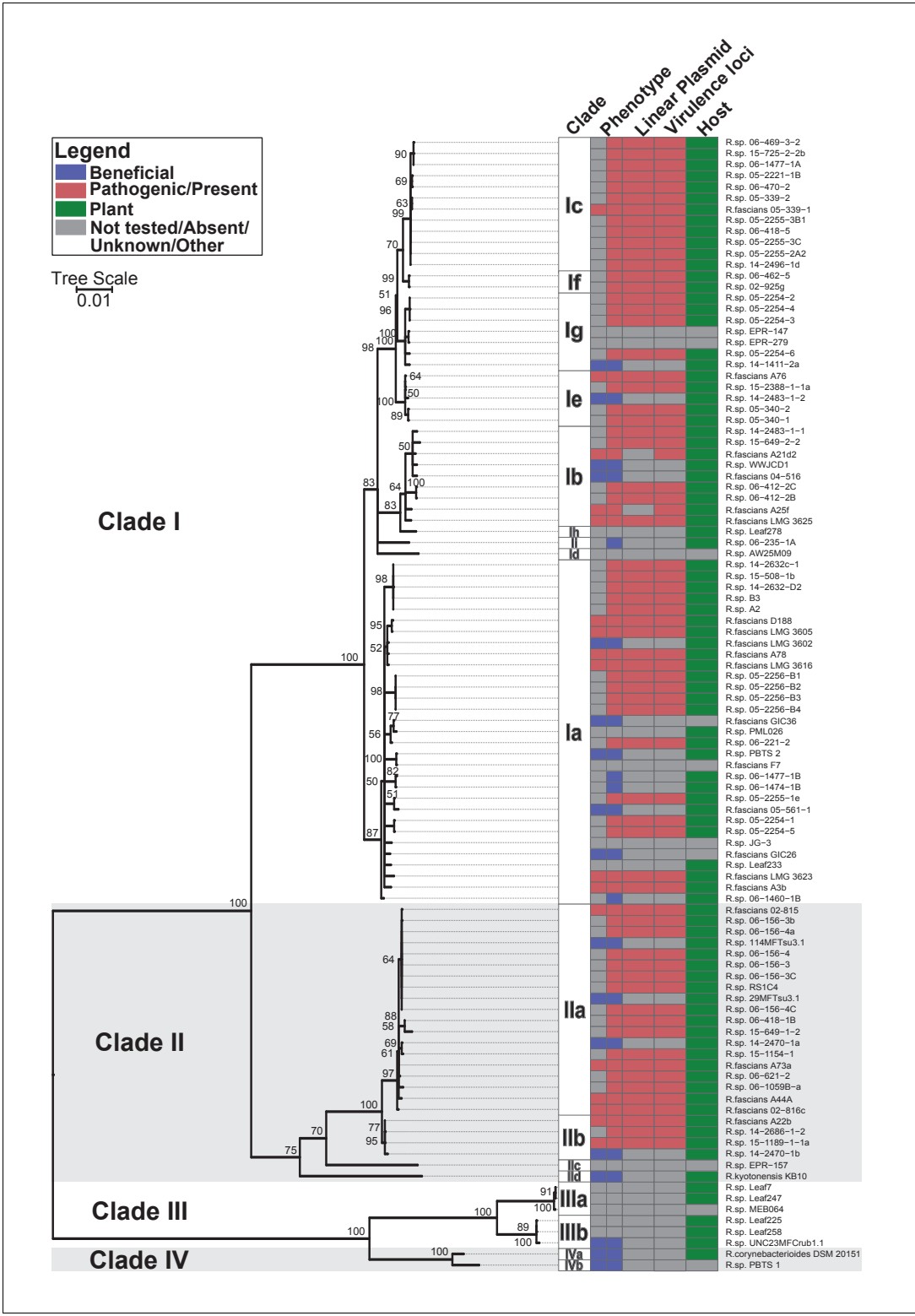

**Figure 1.** Plant-associated isolates of *Rhodococcus* form four sister clades. Multi-locus sequence analysis maximum likelihood tree of plant-associated isolates of *Rhodococcus*. Translated sequences for *ftsY*, *infB*, *rpoB*, *rsmA*, *secY*, *tsaD*, and *ychF* from 104 members of *Rhodococcus* were identified using TBLASTN, aligned, and used to generate a multi-locus maximum likelihood tree. Clade designations are based on analysis of average nucleotide identity (***Supplementary file 1B***). Columns indicate the features of the corresponding isolate. Grey bars indicate not tested, absent, unknown, or other for phenotype, linear plasmid, virulence loci, and host

*Figure 1 continued on next page*

*Figure 1 continued*
columns, respectively. The left-half of the column corresponding to phenotype indicates a confirmed phenotype, whereas the right-half indicates an inference based on the presence or absence of virulence genes.
DOI: https://doi.org/10.7554/eLife.30925.003
The following figure supplement is available for figure 1:

**Figure supplement 1.** Plant-associated *Rhodococcus* form a distinct clade.
DOI: https://doi.org/10.7554/eLife.30925.004

Fifty-one of the newly sequenced isolates encode virulence genes. We inspected their genome assemblies as well as those from previously sequenced pathogenic isolates (*Creason et al., 2014b*). All but four of the 66 genome assemblies had *att*, *fasR*, and *fas* on the same contig as pFi_009. The *fas* locus is present in a region that is conserved in pFiD188, the virulence plasmid of isolate D188, suggested to be necessary for plasmid replication and maintenance (*Francis et al., 2012*). The pFi_009 gene is predicted to encode a telomere-associated protein hypothesized to be necessary for the replication of the linear virulence plasmid. The genome assemblies of isolates 06-469-3-2 and 05-2254-6 had contig breaks that disrupted the linkages between virulence and plasmid-associated loci, but these contigs were nonetheless similar in composition and are co-linear to the reference plasmid sequence. The average sequencing coverage of plasmids relative to that of corresponding chromosomes was 1.89 ± 0.56. Only three assemblies had coverages less than 1.0 but all three had virulence loci on the same contig as pFi_009. A21d2 and A25f, previously sequenced, are exceptional because the virulence loci are encoded in their chromosomes (*Creason et al., 2014b*). Therefore, of the 66 isolates confirmed or inferred to be pathogenic, 64 carry a virulence plasmid. Pathogenic isolates were assigned to eight different species (*Figure 1*).

We identified single nucleotide polymorphisms (SNPs) for 82 isolates and used these SNPs to define genotypes and to assemble two clade-specific minimum spanning networks (*Supplementary files 1C and 1D*; *Figure 2*). The genotypes show pairwise differences in between 220 and 11,714 SNPs. We mapped nursery information onto the network, which provided information on potential transmission patterns (*Figure 2*). In this figure, the 'a' identification is associated with nurseries in which we identified evidence of multiple and independent infections. Plants from nursery N15, indicated with 'a1', were infected by five genotypes belonging to Clade I and two belonging to Clade II. Nursery N8 ('a2') and several others were also associated with multiple genotypes. However, nursery N8 had a single host plant that was infected by at least three genotypes that represented six of the cultured isolates. The three isolates within one of these genotypes differ by up to 20 pairwise SNPs, whereas the two isolates in one of the other genotypes have no differences (*Supplementary file 1C*). The third genotype is separated from the other two by 231 and 1414 pairwise SNPs (*Figure 2E*). Epidemiological link 'b' was detected in nursery N15. The corresponding genotype includes seven isolates from Clade IIa, sampled four years apart from *Campanula* plants (*Supplementary file 1A*). These isolates are separated by 0–2 pairwise SNPs. Epidemiological links designated as 'c' were also made between isolates collected from geographically separated nurseries. Nurseries N1, N12, and N13 ('c1') had isolates of the associated genotype (Clade IIb) that are separated by 1–4 pairwise SNPs and were collected up to 13 years apart from *Leucanthemum* and *Geranium* plants. Nurseries N7, N11, and N14 ('c2') had isolates of the associated genotype (Clade Ia) that were separated by 12–20 pairwise SNPs. The isolates were collected nine years apart, and from *Veronica* plants.

## Multiple distribution patterns of plasmids occur in isolates cultured from agricultural settings

The virulence plasmids of 64 plant pathogenic *Rhodococcus* isolates were categorized on the basis of the phylogenetic analysis of 123 genes that are present in at least 95% of the virulence plasmids, and sub-categorized on the basis of patterns of gene presence/absence (*Figure 3*; *Figure 3—figure supplement 1*; *Supplementary file 1E*). The phylogeny has strong support for two major plasmid types, as well as one unique type only carried by isolate LMG3616 (*Figure 3—figure supplement 1*). The plasmid sequences within the major clades are conserved and provided few informative nucleotide polymorphisms for sub-categorizing plasmids. The greatest contribution to plasmid diversity is

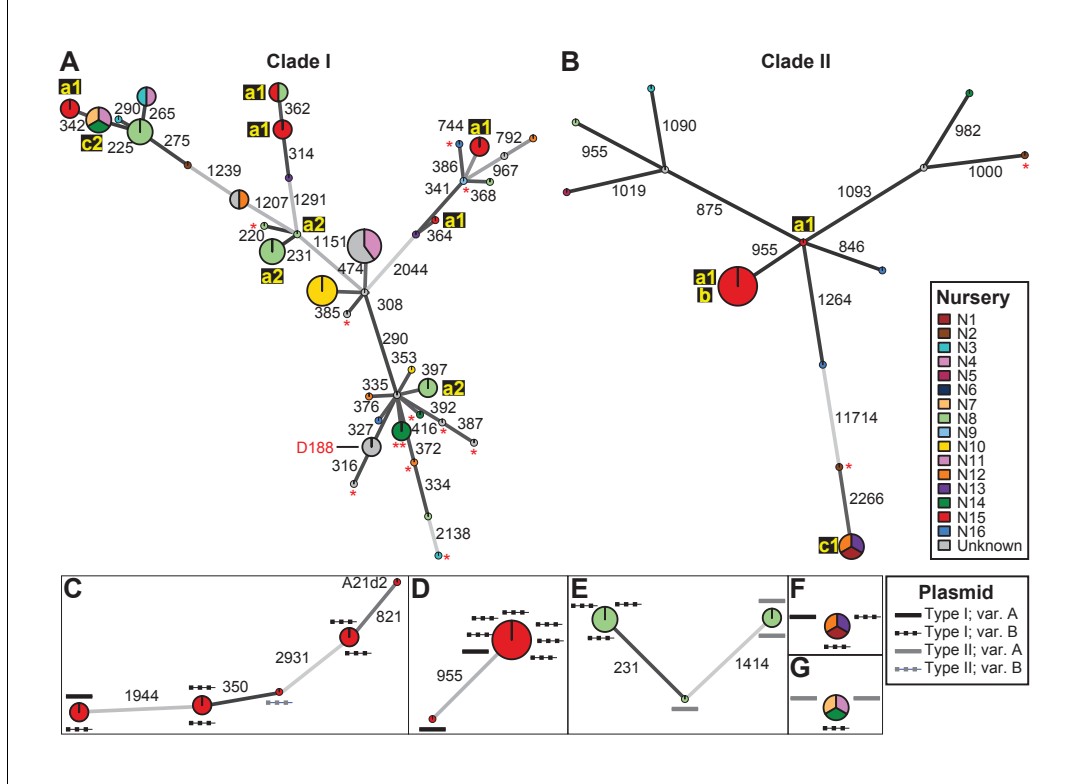

**Figure 2.** Analysis of SNPs reveals three transmission patterns of pathogenic *Rhodococcus*. Minimum spanning networks of isolates of (**A**) Clade I and (**B**) Clade II. Each genotype is displayed as a circle, with sizes scaled to represent the number of associated isolates (smallest = 1 isolate). Colors represent the source of the isolates (see key), with coloring proportional to the ratio of isolates from each source. Lower-case letters and numbers (a1, a2, b, c1, and c2) highlight potential transmission patterns; see panels **C–G**. Asterisks = lacking virulence genes. The genotype that includes D188 is indicated. (**C, D**) Minimum spanning networks of pathogenic isolates belonging to Clade I (**C**; 'a1') and Clade II (**D**; 'a1' and 'b') from nursery N15. A21d2 lacks a virulence plasmid and its virulence loci are present in the chromosome. (**E**) The minimum spanning network for isolates of pathogenic isolates from nursery N8 ('a2'). (**F**) The epidemiological link 'c1' between isolates from nurseries N1, N12, and N13. (**G**) The epidemiological link 'c2' between isolates from nurseries N7, N11, and N14. Plasmid types and their variants are mapped onto each of the nodes (see key). Numbers adjacent to connecting lines indicate the number of SNPs that separate each genotype. The lengths of connecting lines are arbitrary; gray lines indicate distances that exceed an arbitrary threshold.

DOI: https://doi.org/10.7554/eLife.30925.005

gene gain and loss, and the relative clustering of plasmids based on the presence/absence of genes allowed subgrouping into INDEL variants, as indicated with uppercase letters *in Figure 3*. At the level of plasmid type, presence/absence categorization was identical to the phylogeny (*Figure 3— figure supplement 1*). The genes that define each INDEL variant are often present in largely contiguous regions in the plasmid, and could have been acquired as blocks (*Francis et al., 2012*).

We next characterized the distribution of plasmids in epidemiologically linked isolates and showed that the patterns of plasmids were inconsistent with those expected of transmission between isolates of a chromosomal genotype (*Figures 2C–G and 3*). One genotype associated with 'a1' (the most left node of network; *Figure 2C*) includes isolates 05-339-1 and 05-339-2, which differ by seven pairwise SNPs and which carry Type IA and Type IB plasmids, respectively. The genotype associated with 'b' includes isolate 02–815, which carries a Type IA plasmid (*Figure 2D*). This isolate is epidemiologically linked to six isolates (0–2 pairwise SNPs), but the isolates collected four years later carry a Type IB virulence plasmid. Likewise, the genotype associated with 'c1' includes isolate A22b, which carries a Type IA virulence plasmid and is linked to two isolates carrying Type IB plasmids (*Figure 2F*). The most striking deviation is found in the genotype associated with 'c2' (*Figure 2G*). Isolate 06-1477-1A carries a Type IB plasmid and is epidemiologically linked to two isolates that carry a Type IIA plasmid. Even the two Type IIA plasmids are dissimilar as one has a block of genes that, along with other gene INDELs, distinguishes it from the other.

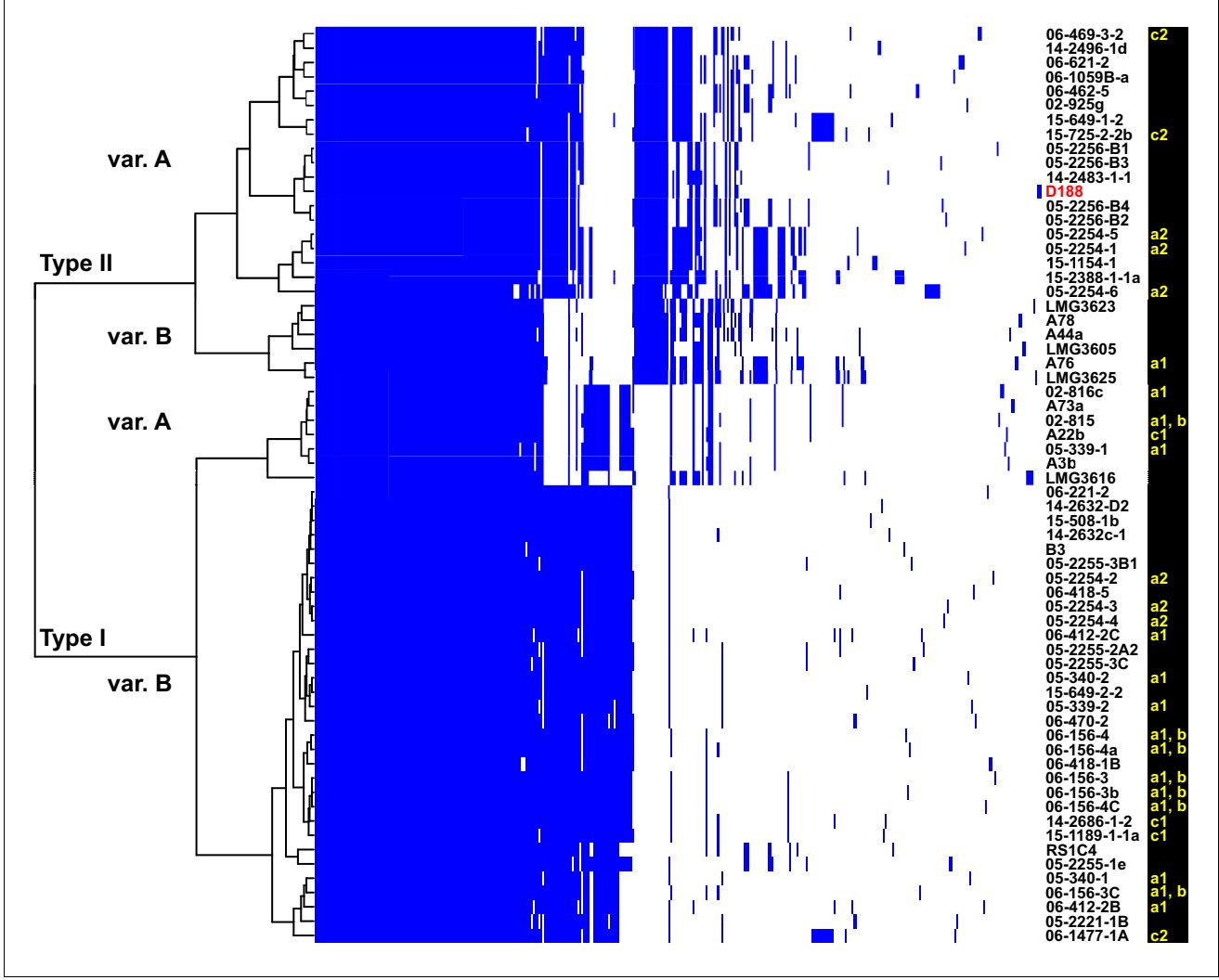

**Figure 3.** Analysis of plasmid variation reveals multiple patterns in distribution. Rows indicate genes present in (blue) or absent from (white) the plasmids of isolates (listed to the right; D188 is labeled in red for reference). Columns represent individual genes. Type categories were determined on the basis of phylogenetic analysis of the core genes. INDEL variants, delineated by gray and white shading, were determined on the basis of the cladogram. The plasmid carried by LMG3616 is enclosed by dotted lines. The lower-case letters and numbers (a1, a2, b, c1, and c2) listed along the right, relate isolates and their plasmids to the potential transmission patterns indicated in *Figure 2*.

DOI: https://doi.org/10.7554/eLife.30925.006

The following figure supplements are available for figure 3:

**Figure supplement 1.** Presence/absence-cladogram and phylogeny support the classification scheme for the virulence plasmid.

DOI: https://doi.org/10.7554/eLife.30925.007

**Figure supplement 2.** Both types of virulence plasmids are present in both clades of pathogenic *Rhodococcus*.

DOI: https://doi.org/10.7554/eLife.30925.008

The diverse genotypes of *Rhodococcus* isolated from nurseries can carry similar plasmid types and these types are not taxonomically restricted. Isolates 06-412-2B of Clade Ib, 05-340-1 of Clade Ie and 06-156-3c of Clade IIa, all of group 'a1' detected in nursery N15, have Type IB plasmids that differ by only a few gene INDELs (*Figure 2C and D*). The phylogeny shows that there are cases in which isolates from the same clade, such as RS1C4 and 06-1059B-a (solid black lines), carry different types of plasmids (*Figure 3—figure supplement 2*; solid black lines). It was also observed that distantly related *Rhodococcus* isolates carry the same type of plasmid. For example, 15-649-1-2 of Clade II and LMG 3623 of Clade I both carry Type II plasmids (*Figure 3—figure supplement 2*; solid red lines).

Two pathogenic isolates were excluded from the analysis because the virulence loci are present in their chromosomes (*Creason et al., 2014b*). A25f was recovered from nursery N12, whereas A21d2 was recovered from nursery N15 (*Supplementary file 1A*).

## *Rhodococcus* isolates that lack the virulence gene promote changes in root architecture

In this dataset, we identified nine isolates of *Rhodococcus* that lack virulence genes. We also identified five such isolates in a previous study, and often fail to detect virulence genes while diagnosing *Rhodococcus* cultured from diseased plants (*Creason et al., 2014b*). Others have implied that these are strains that have lost the plasmid (*Nikolaeva et al., 2009*). The genetic diversity of co-existing isolates described here suggests otherwise (*Figure 1*; *Supplementary files 1A-D*). Pathogenic 06-1477-1A and virulence-gene-lacking 06-1477-1B were isolated from a symptomatic *Veronica* plant, belong to Clades Id and Ia, respectively, and have 1521 pairwise SNPs. Pathogenic isolate 14-2483-1-1 (Clade Ib) was cultured from the same symptomatic plant as virulence-gene-lacking isolate 14-2483-1-2 (Clade Ie), and the two differ by 2820 SNPs. Interestingly, 14-2483-1-2 has the *attR* and *attX* virulence genes, as well as 228 nucleotides of the *attA* virulence gene, on a contig that is dissimilar in sequence and greater in length than the virulence plasmids. The first two coding sequences and the intergenic regions have ≥88% nucleotide identity to corresponding sequences in D188. For *attA*, only the first 65 nucleotides are identical to its homolog in D188. Results from PCR confirmed that the structure of the locus was not a result of misassembly. Two virulence-gene-lacking isolates, 14-2470-1a and 14-2470-1b, were cultured from a symptomatic plant. These two isolates are in Clades IIa and IIb, respectively, and differ by 13,858 SNPs. Of the isolates cultured and tested from this plant, no pathogenic isolate was detected.

An alternative explanation for the presence of genetically diverse, virulence-gene-lacking isolates of *Rhodococcus* is that such isolates are beneficial and enriched for by plants. We therefore compared the symbiosis phenotype of seven virulence-plasmid-lacking isolates that represent the four clades against four virulence-gene-carrying isolates, D188, A44a, A25f, and A21d2 (*Supplementary file 1F*; *Creason et al., 2014b*; *Desomer et al., 1988*; *Lundberg et al., 2012*; *Miteva et al., 2004*). The virulence-gene-carrying isolates were previously determined to be pathogenic, and were selected on the basis of having variations in the structure and sequence of their virulence loci (*Creason et al., 2014b*). PBTS1 and PBTS2, implicated as outbreak strains and cultured from the leaf endophytic compartment of pistachio, were also included (*Stamler et al., 2015b*).

The seven virulence-gene-lacking isolates, as well as PBTS1 and PBTS2, failed to cause disease when inoculated onto the meristems of mature *Nicotiana benthamiana*. The four pathogenic isolates caused leafy galls (*Figure 4A*). These four were also the only ones tested that caused significant inhibition of root elongation, thickening of the stem, and terminal arrest at the cotyledon stage (no primary growth or development of lateral roots), when assayed on seedlings of *N. benthamiana* (*Figure 4B and C*). We examined plants up to two months after inoculation, and seedlings remained terminally arrested. Most isolates reduced the vertical growth of roots, compared to that of mock-inoculated plants, but inhibition by virulence-gene-lacking isolates was more variable and not as severe as that measured in the roots of pathogen-inoculated seedlings (p-values were <0.0001 except for the treatment with PBTS2 [p-value = 0.1153]). Importantly, seedlings inoculated with virulence-gene-lacking isolates did not show thickening of the stem or terminal arrest at the cotyledon stage, morphological changes associated with disease (*Figure 4*).

Instead, we noticed that all virulence-gene-lacking isolates caused changes to the architecture of the roots (*Figure 4B*). Relative to mock-inoculated plants, there were proliferations in root hairs and the plants had more lateral roots or earlier development of lateral roots. The former change was quantified in seedlings that were inoculated with members from a subset of the virulence-gene-lacking isolates. There were significant increases in the number of root hairs (averages ranging from 166.1 to 217.3; p-values were all ≤0.0045), compared to mock-inoculated plants (average of 89.8; *Figure 5A and B*). The isolates varied in their effect, with PBTS2 having the strongest measurable effect (217.3; p-value<0.0001). GIC26 provoked the most visually striking proliferation of root hairs and its extreme effect challenged our ability to count and measure root hairs accurately (*Figure 5A*; p-value=0.0003). The average length of the root hairs was significantly longer following inoculation with the four isolates of *Rhodococcus* (averages ranged from 0.02286 to 0.03604 mm vs 0.01186

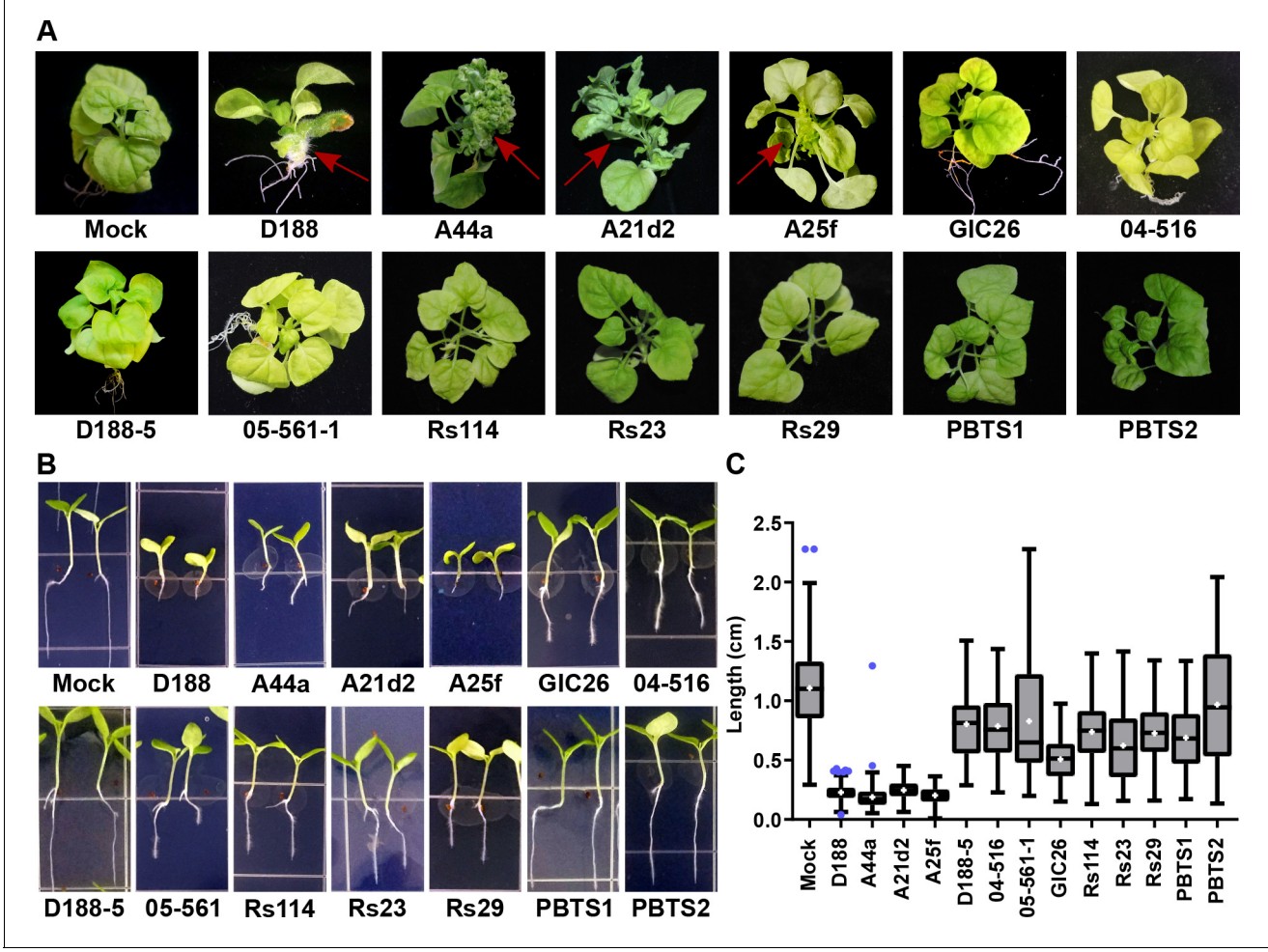

**Figure 4.** The *att, fasR,* and *fas* loci are necessary for the pathogenicity of *Rhodococcus*. (**A**) Representative images of leafy galls on *N. benthamiana*. Isolates of *Rhodococcus* were inoculated at the apical meristem. Red arrows indicate the leafy galls. Rs114, Rs23, and Rs29 are abbreviations for isolates 114MFTsu3.1, UNC23MFCrub1.1, and 29MFTsu3.1, respectively. (**B**) Representative images of the root length of seedlings. Three-day-old *N. benthamiana* seedlings were inoculated with theindicated isolate of *Rhodococcus* or water (mock) and grown vertically for seven days under constant light. Isolates D188, A44a, A21d2, and A25f are the only isolates with virulence genes .(**C**) Quantification of seedling root length. All treatments, except for PBTS2, were significant compared to the mock treatment.

DOI: https://doi.org/10.7554/eLife.30925.009

The following source data is available for figure 4:

**Source data 1.** Lengths of *N. benthamiana* seedling roots 7 days after inoculation with wild type *Rhodococcus*isolates.

DOI: https://doi.org/10.7554/eLife.30925.010

mm in mock-inoculated plants; p-values were all <0.0001; *Figure 5C*). Pathogenic isolate D188 was included as a control, but the root hairs of plants that were inoculated with this isolate were too sparse in number to warrant quantification (*Figure 5A*). Five additional isolates, including 14-2483-1-2 which has part of the *att* locus, were tested and shown to cause changes to the architecture of the roots of plants (*Figure 5—figure supplement 1*).

We analyzed the genome sequences to identify genes that are potentially involved in providing beneficial traits (*Figure 5—figure supplement 2*; *Bruto et al., 2014*; *Glick, 2012*; *Sparacino-Watkins et al., 2014*). Few homologs or pathways were associated with beneficial traits or an endophytic lifestyle. There was also no obvious correlation with symbiosis phenotype. Some isolates have a homolog predicted to encode 1-aminocyclopropane-1-carboxylate (ACC) deaminase, but this sequence had a strong phylogenetic signal, and is predominantly in members of Clade Ia (*Glick, 2014*). No complete tryptophan-dependent auxin biosynthetic pathway was identified

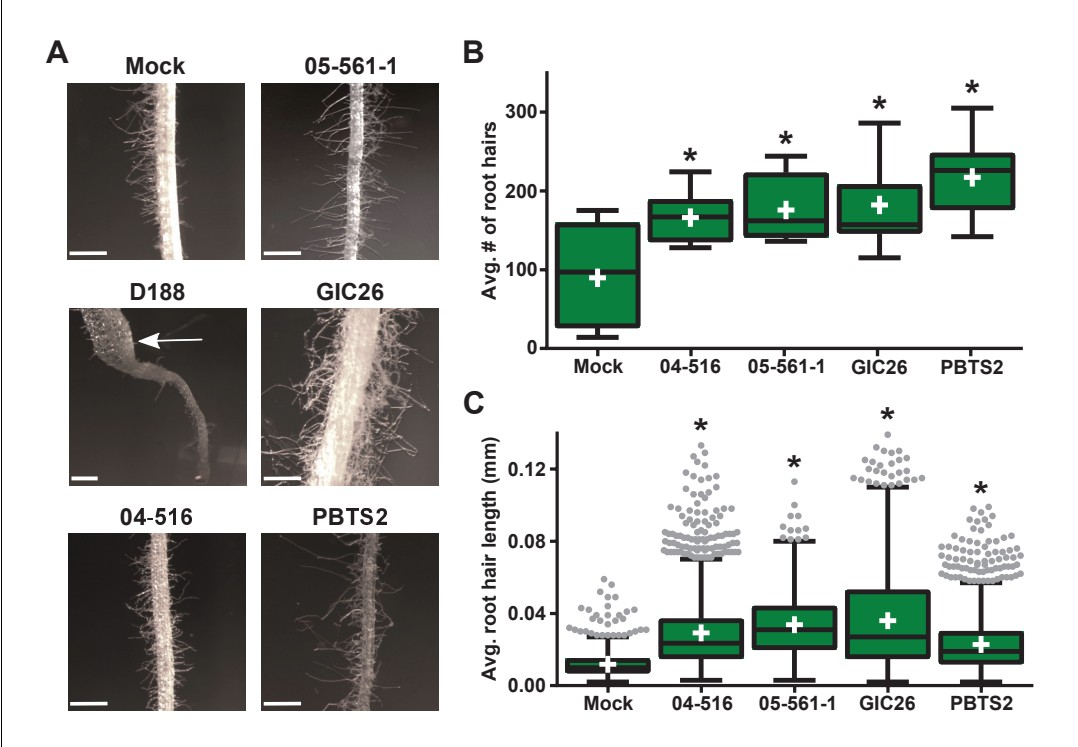

**Figure 5.** Plant-associated *Rhodococcus* bacteria cause changes to the root architecture of seedlings. (**A**) Representative images of root hairs of *N. benthamiana* that were inoculated with isolates of *Rhodococcus*. Images were taken 25 days post inoculation (dpi). The white arrow indicates the thicker stem induced only by isolate D188. Scale bars are 0.5 mm. (**B**) Quantification of average root hair number at 25 dpi. All root hairs were manually counted for at least five seedlings per treatment. (**C**) Quantification of root hair lengths at 25 dpi. All root hairs were manually measured for at least five seedlings per treatment. For **B** and **C**, data were repeated in two independent biological replicates. * indicates a significant difference compared to the mock treatment.

DOI: https://doi.org/10.7554/eLife.30925.011

The following source data and figure supplements are available for figure 5:

**Source data 1.** Numbers of *N. benthamiana* seedling root hairs 25 days after inoculation with wild type *Rhodococcus* isolates.
DOI: https://doi.org/10.7554/eLife.30925.012
**Source data 2.** Lengths of *N. benthamianaroot* hairs 25 days after inoculation with wild type *Rhodococcus* isolates.
DOI: https://doi.org/10.7554/eLife.30925.013
**Figure supplement 1.** Five additional virulence-gene-lacking isolates of *Rhodococcus* cause changes to root architecture.
DOI: https://doi.org/10.7554/eLife.30925.014
**Figure supplement 2.** Heatmap of genes and functions enriched in clades of plant-associated *Rhodococcus*.
DOI: https://doi.org/10.7554/eLife.30925.015

(*Spaepen et al., 2007*). The most highly represented classes of carbohydrate active enzymes (CAZYmes) are members of carbohydrate esterase groups CE1 and CE10. CE1 includes xylanases, which degrade hemicellulose (*Lombard et al., 2014*). CE10 consists of cholinoesterases, a group of enzymes that act on non-carbohydrate sources. In addition, we identified cutinases (C5) and pectate lyases (PL22). These classes of enzymes may contribute to the endophytic lifestyle of *Rhodococcus*. There are 108 genes that are enriched in the genomes of isolates from the four plant-associated clades (*Supplementary file 1G*). Forty genes are annotated as hypothetical and many others lack sufficient information in their annotations. AntiSMASH analysis identified anywhere from 1 to ~ 20 loci that may be involved in the production of secondary metabolites (*Figure 5—figure supplement 2*; *Blin et al., 2017*). Few had sufficient similarities to previously characterized loci to allow the inference of the identity of the metabolite. Whether and how the functions of these genes contribute to the plant-associated lifestyle are unknown.

## The virulence plasmid is sufficient to transition *Rhodococcus* from being potentially beneficial to being pathogenic

The pFiD188Δ*att* virulence plasmid was successfully conjugated into a subset of the *Rhodococcus* isolates that originally lacked virulence genes. This plasmid variant encodes *fasR* and *fasA-F*, but has a kanamycin resistance gene that disrupts *attR*, *attX*, and *attA-G* (*Maes et al., 2001*). Regardless, plasmid pFiD188Δ*att* is sufficient for isolate D188 to cause disease in mature plants and seedlings, and plants that were treated with a strain containing this plasmid were no different from those infected with D188 (p-value>0.9999; *Figure 6—figure supplement 1*; *Maes et al., 2001*). Despite repeated attempts, we were not able to conjugate the plasmid into isolates outside of Clade I successfully. Each of the pFiD188Δ*att*-carrying isolates caused leafy galls on plants (*Figure 6A*). In addition, these isolates were no longer capable of causing increases in the growth of root hairs, unlike their corresponding near-isogenic genotypes (*Figure 6B*). Instead, the isolates carrying pFiD188Δ*att*, when compared to their near-isogenic plasmid-lacking genotypes, caused disease symptoms and significantly inhibited the growth of seedlings (*Figure 6B–C*; p-values were all <0.0001).

The inverse transition was also demonstrated. We isolated a variant of D188, D188ΔpFiD188, which lacks the virulence plasmid. When inoculated onto roots of *N. benthamiana* seedlings, D188ΔpFiD188 caused changes to the architecture of root systems and no longer caused disease to mature plants (*Figure 6B*; *Figure 6—figure supplement 2*). We had to generate a new plasmid-lacking strain because the previously generated D188-5 is compromised in in vitro growth (*Figure 1*; *Figure 6C*; *Figure 6—figure supplement 2*; *Desomer et al., 1988*). Analysis of its genome sequence revealed a significant deletion of 25.4 kb from the chromosome (*Supplementary file 1H*). Most of the affected 25 genes have annotated functions implicated in housekeeping functions. Sequencing of D188ΔpFiD188 confirmed that it only lacked the virulence plasmid. This isolate also grew similarly to D188 and had no measurable fitness defects (*Figure 6C*; *Figure 6—figure supplement 2*).

Only three loci on pFiD188 have been implicated in virulence. We have not been able to repeat results showing that the deletion mutant of *att* is attenuated in virulence (*Figure 6—figure supplement 1*; *Crespi et al., 1992*; *Maes et al., 2001*). But when constitutively expressing *attR*, a homolog of the LysR transcriptional regulator necessary for *att* gene expression, D188 caused unusual leafy galls on *N. benthamiana* (*Figure 6—figure supplement 3*). Unlike normal galls that terminate primary growth, those caused by the *attR*-overexpressing strain regained meristematic activity. When inoculated onto roots, the symptoms were more variable, but nonetheless similar to those caused by D188. The effects were significantly different relative to those seen in mock-inoculated seedlings (p-value<0.0001) or in those inoculated with D188 (p-value<0.0001). The *fas* locus is predicted to be necessary for *Rhodococcus* to produce and secrete a mix of cytokinins (*Pertry et al., 2009*). Approximately 0.1 μM of the synthetic cytokinin 6-benzylaminopurine (BA) was equivalent to a starting inoculum of only ~$2.5 \times 10^3$ colony-forming units (cfu) of D188 (*Figure 7A*). However, regardless of the amount of BA in the medium, the exogenously applied cytokinins only inhibited root elongation and did not provoke the thickening of stems or arrest plant growth.

## PBTS1 and PBTS2 are not outbreak strains

Our results show that PBTS1 and PBTS2 are not pathogenic on *N. benthamiana* (*Figures 4* and *5*). We next tested whether altering the dose influences the outcome of interaction between *N. benthamiana* and PBTS2. As inoculum levels of PBTS2 were increased, there was a greater reduction in root length (*Figure 7B*), but the effect was never to the same robustness and degree as that measured in seedlings infected with D188. In addition, PBTS2 did not cause thickening of stems or terminal arrest in the growth of the plant. At 28 days post-inoculation (dpi), the leaves of seedlings inoculated with the highest tested levels of PBTS2 had developed to the same stage as those of mock-inoculated seedlings, whereas D188-inoculated seedlings remained arrested in growth (*Figure 7C*). The roots of seedlings inoculated with PBTS2 also formed lateral roots. As inoculum levels of PBTS2 were decreased, there was less reduction of root length, and at the lowest dose tested, roots were significantly longer (1.161 cm; p-value = 0.0043) than those of mock-treated plants (1.007 cm; *Figure 7B*). A pathogenic PBTS2 strain carrying pFiD188Δ*att* showed a dose effect similar to that seen for D188 (*Figure 7B*; p-values were all >0.3584 for all within-dose comparisons).

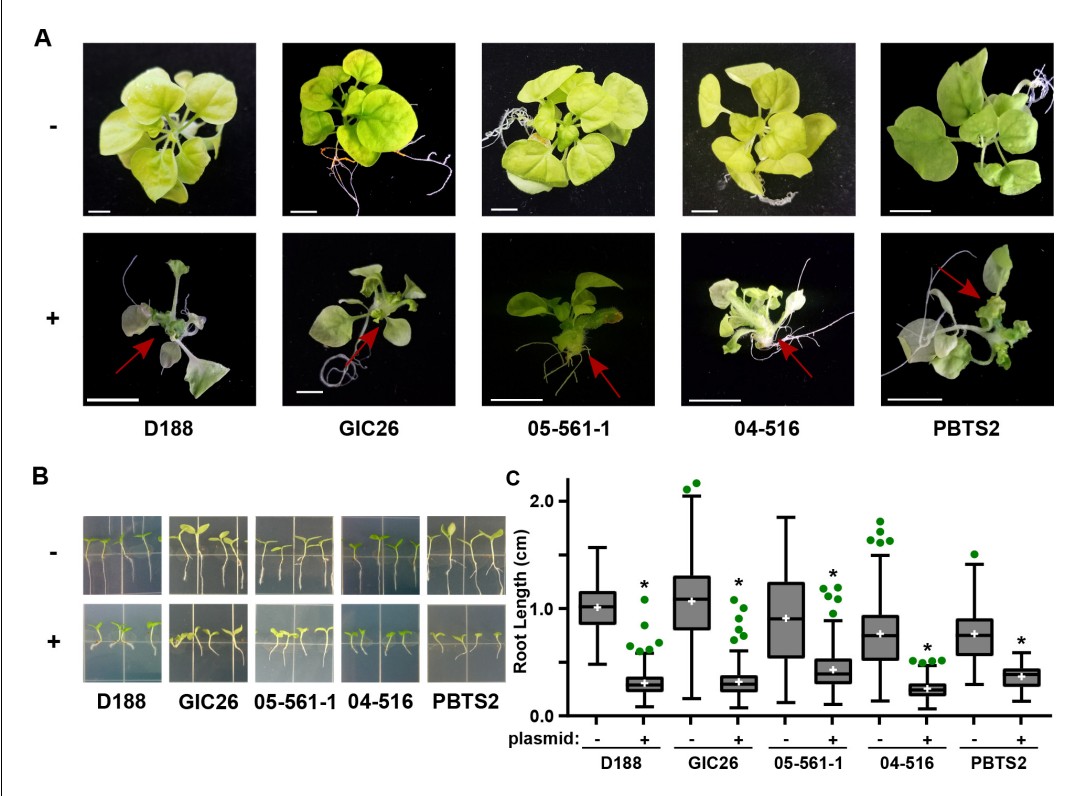

**Figure 6.** Plasmid pFiD188 with functional *fasR and fas* is sufficient to transition *Rhodococcus* isolates to phytopathogens. (A) Representative images of leafy galls on *N. benthamiana*. Red arrows indicate leafy galls. Images for GIC26, 04–516, and PBTS2 are repeated from *Figure 1*. (B) Representative images of the root lengths of seedlings. Three-day-old *N. benthamiana* seedlings were inoculated with the indicated isolate of *Rhodococcus* or water (mock) and grown vertically for seven days under constant light. (C) Quantification of the root lengths of *N. benthamiana* seedlings. In all panels, -/+ indicates absence or presence of pFiD188Δ*att*. * indicates a significant difference compared to plants treated with the corresponding genotype lacking the plasmid.

DOI: https://doi.org/10.7554/eLife.30925.016

The following source data and figure supplements are available for figure 6:

**Source data 1.** Lengths of *N. benthamiana* seedling roots 7 days after inoculation with *Rhodococcus* isolates +/- pFiD188Δ*att*.
DOI: https://doi.org/10.7554/eLife.30925.022

**Figure supplement 1.** *Rhodococcus* isolate D188 with plasmid pFiD188Δ*att* is pathogenic.
DOI: https://doi.org/10.7554/eLife.30925.017

**Figure supplement 2.** Eviction of the virulence plasmid reverts isolate D188 to a beneficial bacterium.
DOI: https://doi.org/10.7554/eLife.30925.018

**Figure supplement 2—source data 1.** Optical density values of culture-grown bacteria.
DOI: https://doi.org/10.7554/eLife.30925.019

**Figure supplement 3.** *Rhodococcus* isolate D188 carrying L5::*attR* is affected in virulence.
DOI: https://doi.org/10.7554/eLife.30925.020

**Figure supplement 3—source data 1.** Lengths of *N. benthamiana* seedling roots 7 days after inoculation with isolate D188 +/- L5::*att*R.
DOI: https://doi.org/10.7554/eLife.30925.021

To exclude the possibility that these results are due to incompatibility between PBTS1 and PBTS2 and *N. benthamiana*, other species of plants were tested. We used pea, an indicator species for confirming pathogenic *Rhodococcus*, and UCB-1 pistachio, reportedly the host of the epidemic. Both plant species failed to show disease symptoms, regardless of whether isolates were tested individually or in combination (*Figure 7—figure supplement 1*). Nine additional pistachio isolates that lack virulence genes also failed to cause disease. Four isolates, 14-687, 14-688, 14-694, and 14-700, were cultured from asymptomatic pistachio plants while five, SR18, AGD2M, AGD3B, AGD6D,

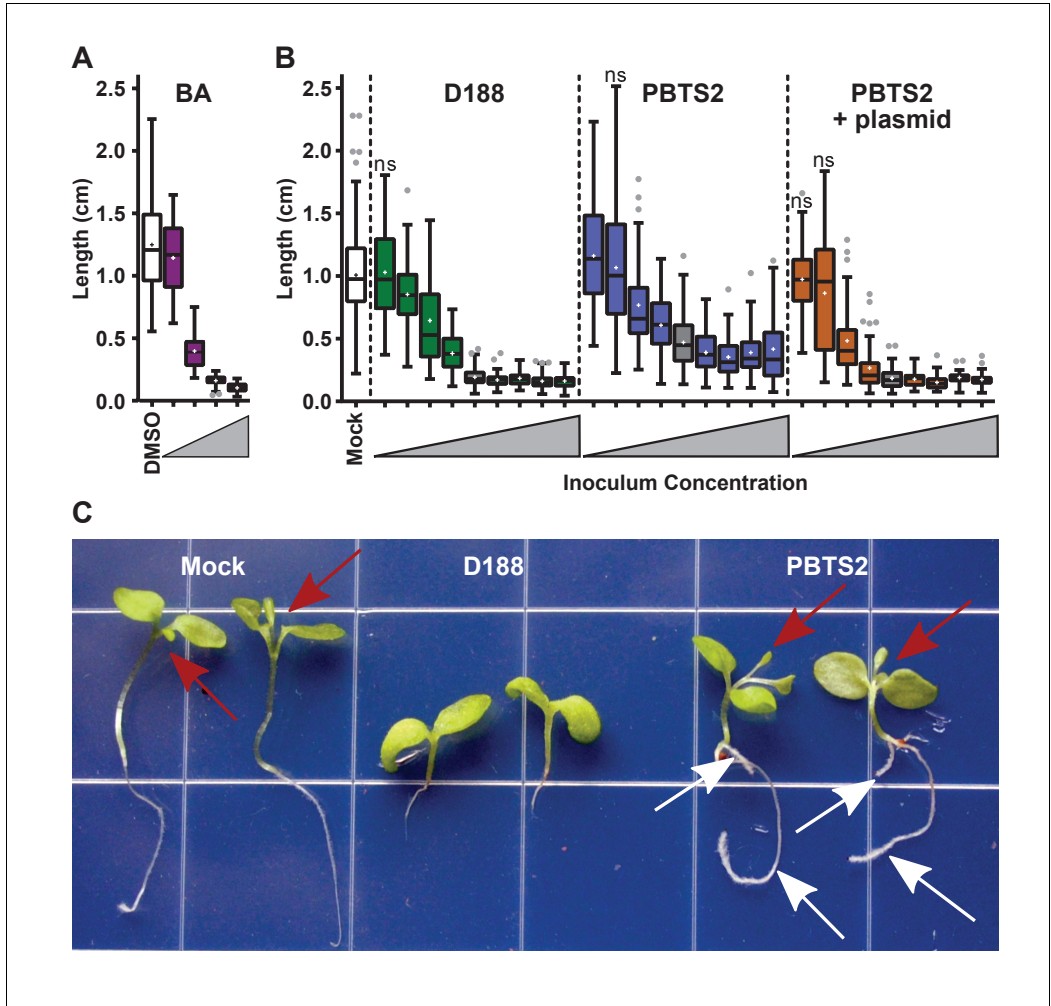

**Figure 7.** *Rhodococcus* has a dose-dependent effect on the root elongation of *N. benthamiana* seedlings. (A) Quantification of the seedling root lengths of plants grown in exogenously applied cytokinin (6-benyzlaminopurine; BA). Three-day-old *N. benthamiana* seedlings were transferred to media supplemented with BA (0.01, 0.1, 1.0, and 10 μM) or dimethyl sulfoxide (DMSO). (B) Quantification of the root lengths of seedlings inoculated with increasing doses of *Rhodococcus*. Three-day-old *N. benthamiana* seedlings were inoculated with isolates D188, PBTS2, or PBTS2 + pFiD188Δatt, with doses ranging from $2.5 \times 10^2$ to $1.0 \times 10^{12}$ colony-forming units (cfu). The sample shaded in gray highlights the inoculum of $OD_{600} = 0.5$ ($1x = 2.5 \times 10^{10}$ cfu) used in all other assays. Inocula below this decrease in 100-fold intervals. Inocula above increase at 2x, 4x, 10x, and 20x. All treatments are significantly different from mock unless otherwise noted with 'ns'. (C) Representative image of morphological changes in seedlings. Seedlings inoculated with *Rhodococcus* D188 or PBTS2 ($5 \times 10^{11}$ cfu; 10x typical amount) or water (mock) were photographed. Red arrows indicate true leaves. White arrows indicate lateral roots and the proliferation of root hairs.

DOI: https://doi.org/10.7554/eLife.30925.023

The following source data and figure supplements are available for figure 7:

**Source data 1.**

DOI: https://doi.org/10.7554/eLife.30925.024

**Figure supplement 1.** *Rhodococcus* isolates PBTS1 and PBTS2 do not cause disease symptoms in tested plant species.

DOI: https://doi.org/10.7554/eLife.30925.025

**Figure supplement 1—source data 1.** Lengths of *N. benthamiana* seedling roots 7 days after growth in BA or inoculation with isolates D188 or PBTS2.

DOI: https://doi.org/10.7554/eLife.30925.026

**Figure supplement 1—source data 2.** Number of stems of peas 14 days after inoculation with isolates D188, PBTS1, or PBTS2.

*Figure 7 continued on next page*

*Figure 7 continued*

DOI: https://doi.org/10.7554/eLife.30925.027

**Figure supplement 1—source data 3.** Length of stems of peas 14 days after inoculation with isolates D188, PBTS1, or PBTS2.

DOI: https://doi.org/10.7554/eLife.30925.028

**Figure supplement 1—source data 4.** Heights of pistachio UCB1 210 days after inoculations with *Rhodococcus* isolates.

DOI: https://doi.org/10.7554/eLife.30925.029

**Figure supplement 1—source data 5.** Internode lengths of pistachio UCB1 210 days after inoculations with *Rhodococcus* isolates.

DOI: https://doi.org/10.7554/eLife.30925.030

and AGD6H, were cultured from symptomatic pistachio plants. Even pathogenic isolates D188 and A44a failed to cause disease symptoms in UCB-1 pistachio.

Another aspect of the diagnosis of pistachio bushy top syndrome was the use of *vicA* to confirm pathogenic *Rhodococcus* (*Stamler et al., 2015a, 2015b*). Primers designed for *fasA* and *fasD* specifically amplified a product of expected size from pathogenic isolates D188, A44a, and A25f, and failed to amplify a product from any of the tested beneficial strains (*Figure 8A*; *Supplementary file 1I*; *Nikolaeva et al., 2012*; *Serdani et al., 2013*). The primers for *fasA*, and *fasD* failed to yield a product from A21d2 because this isolate carries an analog of the *fas* locus (*Creason et al., 2014b*). The molecular detection of *fasR* using a loop-mediated isothermal amplification (LAMP)-based assay specifically distinguished all tested pathogenic isolates from beneficial isolates.

The detection of *vicA* did not follow a pattern consistent with the pathogenicity phenotype (*Figure 8A*). It has a high false-positive rate and detected several, but not all, beneficial isolates. Our repeated attempts to amplify *vicA* from PBTS1 were unsuccessful; there are four and six mismatches between the two primers used for PCR and the *vicA* sequence from PBTS1. Homologs of *vicA* are predicted to be present in nearly all members of the Actinobacteria, including in all 407 *Rhodococcus* isolates for which genome sequences are available. The topologies of the *Rhodococcus* genus and the *vicA* trees are largely congruent, indicating that this locus is mostly vertically inherited, but with some evidence of recombination (*Figure 8B*).

To address the need for on-site molecular tools to distinguish pathogenic from beneficial genotypes, we used a new molecular detection method that is rapid, robust, and sensitive (*Piepenburg et al., 2006*). We targeted *attE* and *attG* because they are the most unique relative to all sequences in the databases and are conserved among the pathogenic isolates that we have examined (*Creason et al., 2014b*). The use of the primers for *attE* and *attG* in standard PCR and recombinase polymerase amplification (RPA) basic successfully amplified products of expected size from DNA of pathogenic strains, including A21d2 and A25f (*Figure 8C*). No products were detected when DNAs from beneficial strains were used as templates. An additional oligonucleotide probe that anneals within the amplified fragment was designed for RPA nfo, and when coupled with modified amplification primers, this probe was successful in detecting a product via lateral flow. This method was specific and discriminated between pathogenic and beneficial *Rhodococcus*. Moreover, RPA nfo can be completed in the absence of specialized equipment, and can yield results in just 30 min.

## Discussion

Whole-genome-enabled epidemiological studies have revealed local, global, and historical patterns for the transmission of human pathogens and have informed on health care (*Comas et al., 2013*; *Croucher et al., 2011*; *Harris et al., 2010, 2013*; *Mutreja et al., 2011*; *Parkhill and Wren, 2011*; *Walker et al., 2013*). Ours is a case study for using genomic epidemiology to uncover and explain the transmission patterns of phytopathogens in agricultural systems. The investigation of plasmid distribution highlighted the significant role of HGT in shaping the population structure of pathogenic bacteria and revealed challenges in modeling their transmission. Our analysis of chromosomal SNPs suggested that nurseries experience multiple and independent introductions of pathogenic *Rhodococcus*, exemplified by 'a' isolates observed in nurseries N15 and N8 (*Figure 2*). The

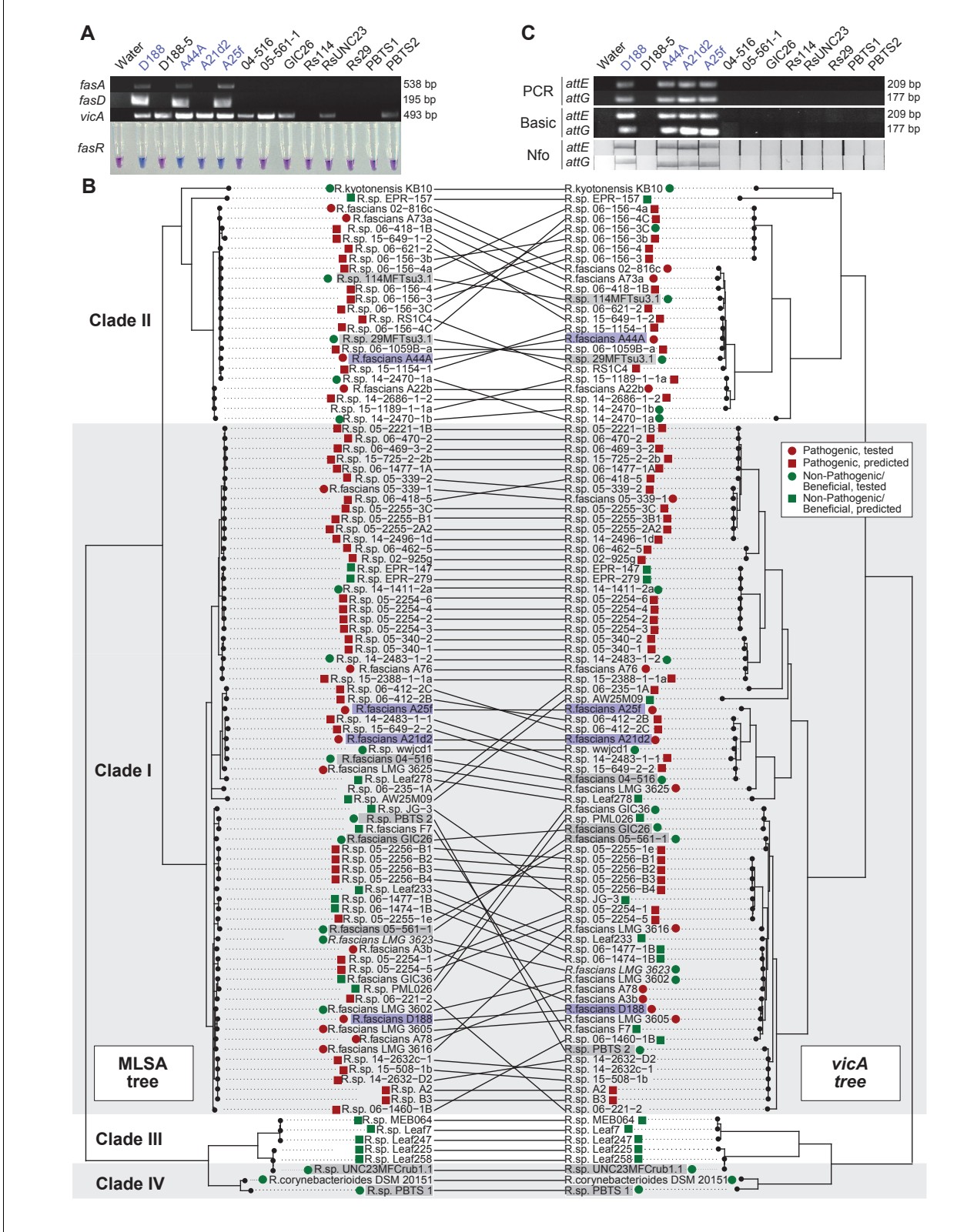

**Figure 8.** Use of the *vicA* gene does not discriminate pathogenic *Rhodococcus*. (A) (Top) Using locus-specific primers and DNA from the listed isolates, fragments of *fasA*, *fasD* and *vicA* were PCR amplified. Pathogenic isolates are labeled in blue; virulence-gene-lacking isolates are labeled in black. Products were resolved on a 1% TAE agarose gel. Amplicon sizes are listed to the right of the gel images. (Bottom) LAMP assay detection of *fasR* from the same DNA samples. A positive result is visualized by a blue color. Negative results are light purple. (B) Congruency between species (left) and *vicA*

*Figure 8 continued*

trees (right). Clades other than I–IV are not shown. Highlighted isolates are the same as in **A**, pathogenic isolates are highlighted in blue, virulence-gene-lacking isolates are highlighted in gray. (**C**) Standard endpoint PCR, RPA Basic, and RPA nfo were used to detect *attE* or *attG* from DNA extracted from isolates of *Rhodococcus*. Pathogenic isolates are labeled in blue. For PCR and RPA basic, product sizes are list to the right of the figure. For RPA nfo, the presence of the test band is indicative of a positive reaction; the control bands for all strips were confirmed (not shown).
DOI: https://doi.org/10.7554/eLife.30925.031

link between isolates collected across time ('b') is indicative of a reservoir population that has a pathogenic genotype. The epidemiological links ('c1' and 'c2') of isolates from different nurseries support the possibility of point source outbreaks and suggest that the sources have reservoir populations.

However, the distribution of plasmids also indicated that alternative processes may be occurring (*Figures 3* and *9*). First, two different plasmid types are associated with 'c2'. This is not expected from an outbreak and is more consistent with different members of a lineage acquiring plasmids separately. Second, different plasmid variants are carried by isolates that are defined by genotypes associated with 'a1', 'b', and 'c1' (*Figure 2*). These are not expected patterns and probably reflect separate acquisitions of plasmids by different members of a lineage or the rapid and independent evolution of plasmids. Third, genetically distinct lineages of *Rhodococcus* at nursery N15 carry the same variant of plasmid. This is best explained by multiple lineages acquiring

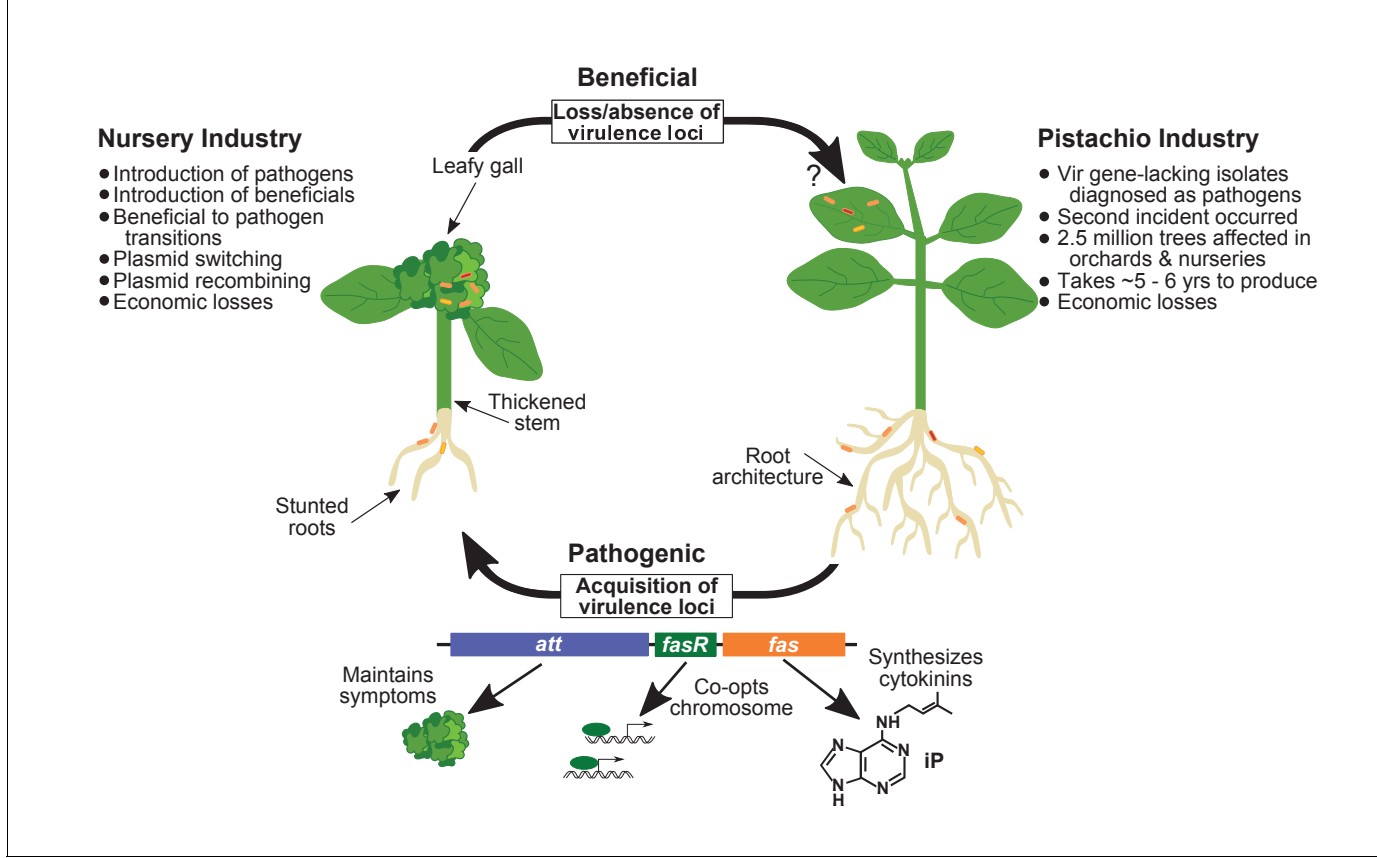

**Figure 9.** Model of the evolutionary transition in *Rhodococcus* and effects on agricultural sectors. The presence or absence of the virulence plasmid in eight species of *Rhodococcus* determines whether the bacterium is beneficial (right), and promotes growth in root architecture, or pathogenic (left), and causes leafy galls and inhibits primary growth. The virulence plasmid carries three loci identified as necessary for pathogenicity and their predicted functions are described. Horizontal gene transfer and evolutionary transitions affect *Rhodococcus* and impact the nursery industry (as indicated on the left of the figure). Virulence-gene-lacking isolates of *Rhodococcus* were diagnosed as outbreak strains and the probable misdiagnosis could have had detrimental impacts on the pistachio industry (as indicated on the right of the figure).
DOI: https://doi.org/10.7554/eLife.30925.032

plasmids from a common donor population. Nurseries often produce a large variety of perennials and clonally propagated plants that are frequently handled and intensely managed in multiple production settings, and are often in regions that produce many different agricultural commodities. These are prime locations for different genotypes of *Rhodococcus* to interact and for plasmids to be transferred, switched, and evolved.

We also demonstrated that plants in agricultural systems are hosts to isolates of *Rhodococcus* that are probably beneficial associative symbionts (*Supplementary file 1A*; *Vacheron et al., 2013*). Root hairs are extensions that increase the surface area of roots, forming an extensive interface between plant and soil. Virulence-gene-lacking isolates of *Rhodococcus* caused significant increases in the number and length of root hairs, which may enable plants to be more efficient in the uptake of water and dissolved nutrients (*Figure 5*). This potentially beneficial growth-promoting phenotype is consistent with the finding of a number of reports identifying *Rhodococcus* within endophytic compartments and the rhizosphere of plants, and with the suggestions that the bacteria are enriched for by plants because of their beneficial traits (*Bai et al., 2015*; *Bodenhausen et al., 2013*; *Bulgarelli et al., 2012*; *Francis et al., 2016*; *Hong et al., 2015, 2016*; *Lundberg et al., 2012*; *Qin et al., 2009, 2011*; *Salam et al., 2017*). The associative symbionts are genetically diverse and represent 14 different species circumscribed by four sister clades (*Figure 1*; *Supplementary file 1B*). The mechanism by which isolates of *Rhodococcus* cause growth changes to plants is unknown, as genome mining efforts suggest that these traits are potentially novel (*Figure 5—figure supplement 2*).

Acquisition of a virulence plasmid by isolates representing eight species of Clades I and II is sufficient to drive an evolutionary transition (*Figures 1* and *6*). Loss of the plasmid reverted *Rhodococcus* to being beneficial, consistent with the hypothesis that virulence genes function irrespective of genomic background (*Figure 6—figure supplement 2*). In addition, we could not identify any chromosome-located genes that are enriched in pathogenic isolates, in comparison to non-pathogenic isolates, that could be potential candidate virulence genes. Evolutionary transitions, such as the switch from being a free-living, non-pathogenic lineage to being a pathogenic lineage, have been detected frequently (for example, by *Bruto et al., 2017*). The transition described for *Rhodococcus* is seamless and the mutualist to pathogen transition has been described only rarely (*Sachs et al., 2011*). It is possible that focus on characterizing binary outcomes in symbioses have obscured the true fluidity of symbioses. Similar transitions may have occurred in *Agrobacterium/Rhizobium*, a group of bacteria that express plasmid-mediated traits of significance to plant agriculture (*de Lajudie et al., 1999*; *Glaeser et al., 2016*; *Hao et al., 2012*; *Lacroix and Citovsky, 2016*; *Wang et al., 2006*).

Transitions and the rapid generation of new lineages of pathogens could occur in agricultural systems, where plants are frequently host to multiple *Rhodococcus* isolates of different genotypes. Pathogenic isolate 05-2254-6, which is associated with 'a2', is most similar to virulence-gene-lacking isolate, 14-1411-2a, also collected from N8. The 220 pairwise SNPs that differ between these two isolates exceeded the threshold used to define genotypes, but the two isolates are nonetheless closely related (*Figures 1* and *2*; *Supplementary file 1C*). This is not unique, as several pathogenic genotypes are related to virulence-gene-lacking genotypes, as indicated by their intermingling in the phylogeny and their connectivity in the network. The patterns involving distinct genotypes or plasmid types, such as 'a1' and 'c2', respectively, are also consistent with evolutionary transitions (*Figures 2* and *3*). An example that is similar to 'a1' is the genotype of D188 and LMG3605, both of which have different plasmid variants.

The limited genetic diversity in virulence plasmids within our dataset was unexpected and suggests that their common ancestor evolved recently (*Figure 3—figure supplement 1*). This is in dramatic contrast to the virulence plasmids of *Agrobacterium* species of bacteria, where the Ti and Ri plasmids form different types that can be easily distinguished on the basis of the phylogeny of a single, conserved virulence gene (*Fuller et al., 2017*). It is also remarkable that the virulence plasmid, which carries only three virulence loci, is sufficient for pathogenicity across eight genetically diverse species of *Rhodococcus* (*Figures 1* and *9*; *Creason et al., 2014b*; *Francis et al., 2012*; *Letek et al., 2010*). FasR is a predicted transcriptional regulator, and is probably key for reprogramming the genome to transition bacteria to pathogens. The roles of the other two loci are still unclear. Results presented here suggest that *att* contributes to the maintenance of disease symptoms, but the mechanism is unknown (*Figure 6—figure supplement 3*). The Fas-produced mix of cytokinins are

predicted to be secreted into plants and necessary to cause disease symptoms, but the existing data are not consistent with this hypothesis (*Pertry et al., 2009*). Exceedingly low amounts of cytokinins are detected in culture-grown bacteria and plants have a variety of cytokinin-buffering mechanisms (*Creason et al., 2014b*; *Kieber and Schaller, 2014*; *Pertry et al., 2009, 2010*). A miniscule amount of starting bacterial inoculum is sufficient to provoke disease symptoms, and exogenous applications of cytokinins fail to phenocopy the effects of pathogenic *Rhodococcus* (*Figure 7*). Last, the cytokinin mixture model is challenged by the revelation that the fitness of D188-5, a key isolate used to develop the model, is severely compromised by a 25-kb deletion (*Figure 6—figure supplement 2*; *Supplementary file 1H*; *Desomer et al., 1988*; *Maes et al., 2001*; *Pertry et al., 2009, 2010*; *Stes et al., 2011*; *Temmerman et al., 2001*).

This study highlights the importance of understanding the genetic and phenotypic characteristics of an organism and the consequences of prematurely drawing conclusions from incomplete data. There is no evidence to suggest that HGT or evolutionary transitions confounded the diagnosis of pistachio, and we could not detect pathogenic *Rhodococcus* from symptomatic tissues of pistachio. We were unsuccessful in repeating results showing that PBTS1 and PBTS2, or other isolates cultured from pistachio, cause disease symptoms on plants (*Figure 4*; *Figure 7—figure supplement 1*). We could not amplify virulence genes from PBTS1 or PBTS2, but when a virulence plasmid was introduced into PBTS2, it was sufficient to transition PBTS2 to a pathogen of a plant species that is demonstrably susceptible to *Rhodococcus* (*Figures 6* and *7B*). Whether pistachio is even a host for pathogenic *Rhodococcus* is unresolved (*Figure 7—figure supplement 1*). Nevertheless, we recognize the insurmountable challenge in showing that there is no possibility that pathogenic *Rhodococcus* causes pistachio bushy top syndrome.

The results from this work prompted us to examine previous studies retrospectively (*Figure 9*; [*Stamler et al., 2015a, 2015b*]). There was a targeted search for *Rhodococcus*, the justification for which is unfounded because the symptoms on pistachio are unlike any produced by pathogenic *Rhodococcus* in any of the 100+ known hosts (*Putnam and Miller, 2007*). Bacteria were inexplicably cultured from asymptomatic leaves distal to symptomatic stem tissues. The key study did not include control strains or reproduce galling and graft failure, the most defining disease symptoms observed in field settings. A high inoculum of *Rhodococcus* was used and high doses of even beneficial bacteria can have costs (*Figure 7*). For example, some human diseases are caused by dysbiosis in which an imbalance of gut microbiotia causes disease (*Bloom et al., 2011*). In addition, hosts often employ mechanisms to regulate or ensure nonpersistent interactions with beneficial bacteria (*Gutjahr and Parniske, 2013*; *Magori et al., 2009*; *Nyholm and McFall-Ngai, 2004*; *Reid et al., 2011*; *Wopereis et al., 2000*).

Results based on molecular detection were similarly tenuous. The use of the *vicA* locus was misleading and led to conclusions regarding the pathogenicity of *Rhodococcus* (*Figure 8*). The reported amplification of fragments of *fasA* and *fasD* (GenBank accession numbers KP274062 and KP274064) from PBTS1 as well as *fasD* (KP274067) from PBTS2 cannot be reconciled with the absence of the genes from the genome sequences (*Stamler et al., 2015b*, *2016*). We used a different technology to re-sequence independently prepared DNA from PBTS2 (*Supplementary file 1F*). The assembly is co-linear with the publically available sequence and lacks the virulence plasmid and virulence genes (*Stamler et al., 2016*). The possibility that virulence plasmids are unstable in populations grown outside of plant environments is not supported by the data. DNA extracted from bacteria grown in culture was used as a template in both PCR and whole-genome sequencing (*Stamler et al., 2015b*, *2016*). We successfully sequenced plasmids from 64 culture-grown pathogenic isolates, which had undergone multiple transfers, and repeatedly and successfully detected pathogenic isolates cultured from symptomatic tissues (*Creason et al., 2014b*).

Some of the results of molecular detection were clearly artifacts. The two fragments reported to correspond to *fasD* are identical in sequence, each consisting of two short fragments of 147 and 194 nucleotides long that are artificially joined together by 280 'Ns'. In our dataset, A21d2 is the only sequenced pathogenic isolate that lacks *fasA*, but it also lacks a homolog of *fasD*. It is thus not expected that PBTS2 should have only *fasD* but no *fasA* (*Stamler et al., 2015b*). The most egregious artifact was the reported amplification of *vicA* from PBTS1 (GenBank accession number KP274063), which we could not reproduce (*Figure 8*). When the sequence of the reportedly amplified fragment was used as a query to BLAST search the PBTS1 genome sequence directly, we failed to identify a homologous region. When used in searches against publically available databases, the top hits other

than KP274063 were NM-J PBTS (KR153287; 100% identity), DMS3-9 (KJ677035; 97% identity), D188 complete genome (CP015235; 96% identity), and PBTS2 complete genome (CP015220; 96% identity).

Another study posited that the virulence plasmids are unstable in populations persisting within plant environments (*Nikolaeva et al., 2009*). Genome sequences showed instead that *Rhodococcus* isolates are from genetically distinct lineages. This observation further emphasizes the need to use appropriate molecular diagnostic tools that discriminate pathogenic from non-pathogenic bacteria. LAMP to detect *fasR* or RPA nfo to detect *attE* or *attG* are both sensitive methods that have the additional benefit of being rapid and less dependent on specialized equipment (*Figure 8*; *Serdani et al., 2013*).

The data are consistent with the possibility that previous conclusions rest on a misdiagnosis of pistachio bushy top syndrome (*Stamler et al., 2015a*, *2015b*). If so, *Rhodococcus* was implicated as pathogenic, irrespective of genotype and in disregard of the genetic and phenotypic diversity of the genus. This mindset conflates all bacteria as causative agents of disease and demotes the importance of bacteria in promoting the health of their hosts (*Figure 9*). This potential misdiagnosis of pistachio bushy top syndrome could be responsible for catastrophic effects. An estimated 2.5 million trees have been destroyed, resulting in tremendous economic loss. Efforts to identify the true cause were decelerated. Accusations regarding the source of *Rhodococcus* have introduced conflict into an industry struggling with a considerable and unfamiliar problem. Considering the data described here, previous conclusions should, at the very least, be tempered and recommendations for managing plants in which *Rhodococcus* bacteria have been detected, reexamined. Ideally, these findings will renew efforts to identify the true nature of the syndrome afflicting UCB-1 pistachio rootstocks.

Actinobacteria are prominent members of plant-associated communities, but the mechanisms and evolution of traits that are important for Gram-positive bacteria to reside in microbial communities and influence plant health are not well understood (*Figure 9*). Members of the *Rhodococcus* genus are excellent models for addressing this knowledge gap. A wealth of associated resources, such as an extensive and diverse collection of genotypes, associated genome sequences, and genetic tools have been developed for *Rhodococcus*. The members of this taxon can also interact with and benefit genetically tractable plant species. Last, the members of *Rhodococcus* are model organisms for characterizing evolutionary transitions between alternative symbiotic states.

## Materials and methods

### Bacterial isolates and growth conditions

*Rhodococcus* isolates used in this study are listed in *Supplementary files 1A and F*. Bacteria were maintained on solid LB medium at 28°C or grown overnight in LB medium at 28°C with shaking. Prior to conjugation, streptomycin-resistant *Rhodococcus* bacteria were selected for each of the recipient genotypes. Conjugations were done as previously described (*Desomer et al., 1988*). Donor and recipient strains were grown in yeast extract buffer (YEB) and shaken at 28°C. Each genotype was mixed at a ratio of 1:1 and filtered through a nitrocellulose filter (pore size, 0.45 μm; diameter, 25 mm; MilliporeSigma, Temecula, CA, USA). The filters were incubated on YEB agar plates for 24 to 28 h at 28°C. The cells were washed from the filter with 5 ml of a buffer containing 10 mM Tris-HCl pH 7.5 and 10 mM $MgSO_4$ and then diluted and plated on YEB medium containing the appropriate antibiotic. *Escherichia coli* was grown on LB medium at 37°C. When appropriate, the medium was amended with 50 μg/ml of antibiotic kanamycin for *Rhodococcus* or *E. coli*. For growth curves, cultures of overnight-grown *Rhodococcus* were pelleted, washed, and resuspended at $OD_{600}$ = 0.5 in a final volume of 200 μl of LB medium in 96-well flat-bottomed plates. The bacteria were grown for a period of 14 hr at 28°C with shaking in a Tecan Spark 10 m plate reader. Optical density ($OD_{600}$) measurements were taken every hour. Three technical replicates were included for each isolate, and the experiment was repeated at least three times with similar results.

### Genome sequencing, assembly, and annotation

The Wizard genomic prep kit (Promega, Fitchburg, WI, USA) was used to extract genomic DNA from *Rhodococcus*. Directions for Gram-positive bacteria were followed. DNA was quantified with a Nanodrop spectrophotometer and adjusted to 50 ng/μl. Total genomic DNA was used to prepare

Nextera XT libraries, and the resulting multiplexed libraries were sequenced on an Illumina HiSeq 3000 to generate 250mer paired end sequencing reads (Center for Genome Research and Biocomputing [CGRB], Oregon State University). Reads were processed as follows. FastQC was used to assess sequencing reads for quality (*Andrews, 2014*). BBduk v.35.82, with the parameters 'ktrim = r k = 23 mink = 9 hdist = 1 minlength = 100 tpe tbo', was used to remove adapter sequences (*Bushnell, 2014*). SPAdes v. 3.1.1, with the parameters '–careful -k 21,33,55,77,99' was used to correct errors and to de novo assemble the reads into contigs (*Bankevich et al., 2012*). Blobtools was used to assess assemblies and guide elimination of contigs likely to be derived from contaminating bacteria (based on combined GC content, coverage, and contig annotation) (*Kumar et al., 2013*). Prokka was used to annotate the assembled genome sequences (*Seemann, 2014*).

## Phylogenetic analyses

Sequences for the maximum likelihood multi-locus sequence analysis (MLSA) tree were acquired using the autoMLSA tool (*Davis Ii et al., 2016*). The sequences for genes, *ftsY* (ABG98302.1), *infH* (ABG98417.1), *rpoB* (ABG93773.1), *rsmA* (ABG97450.1), *secY* (ABG97930.1), *tsaD* (ABG97962.1), and *ychF* (ABG97656.1) from the genome sequence of *Rhodococcus jostii* RHA1 were translated and used as queries in TBLASTN v. 2.2.31 searches against the assembled genome sequences and the NCBI nt database, masked to *Rhodococcus* (*Adékambi et al., 2011*; accessed 12/2016). Of those from NCBI nt, eight strains lacking all seven sequences and/or duplicate results were removed from the analysis. The sequences were aligned using MAFFT v. 6.864b with default settings (*Katoh and Standley, 2013*). A RAxML accessory script was used to determine the best-fitting protein model for each protein sequence alignment. Phylogenetic trees (100 ML searches, 'autoMRE' criterion bootstrap replicates) were generated using RAxML v. 8.1.17 with a partitioned alignment of the MLSA protein sequences (*Stamatakis, 2014*).

A similar analysis using *R. fascians* D188 *vicA* (AMY55488.1) as a query was used to acquire and assemble a phylogeny of 162 malate synthase gene sequences from the NCBI nr database masked to *Rhodococcus* (accessed 04/2017). A cophylo plot of the MLSA and *vicA* trees was generated using the R package phytools (*Revell, 2012*).

For the genes present in 95% of the virulence plasmids, sequences were concatenated using the R package EvobiR SuperMatrix function prior to constructing phylogenies (*Blackmon and Adams, 2015*).

Only bootstrap values greater than 50 are shown.

## Genome analyses and bioinformatics tools

Bowtie2 v. 2.2.3, with the option '–local', was used to align reads to the chromosome references sequences of D188 (CP015235.1) or A44a (GCF_000760735.1), based on the clade assignment of the corresponding isolates (*Langmead et al., 2009*). Alignments were converted to bam format using samtools v. 0.1.18 and read groups were added using Picard tools v. 2.0.1 (*Li et al., 2009*; *Picard Tools, 2015*). GATK v. 3.7 HaplotypeCaller and the options '-ERC GVCF -ploidy 1' were used to call variants for each isolate, and the data were then combined using GenotypeGVCFs (*McKenna et al., 2010*). Variants were filtered using the R package vcfR with depth filtering using quantile probabilities of 0.25 and 0.75 as cutoffs and a minimum of four reads, as well as a missing data cutoff of 20% (*Knaus and Grünwald, 2017*). Variants were converted into a fasta alignment using bcftools v. 1.3–14-ge0890a1 vcf-to-tab and the perl script vcftab-to-fasta (*Li et al., 2009*; *Chen, 2012*). Genotypes were called based on a threshold of 25 SNPs, and bitwise distances, using the R package poppr, were used to assemble minimum spanning networks (*Kamvar et al., 2014*).

Pairwise average nucleotide identity (ANI) between *Rhodococcus* isolates was calculated using autoANI (*Davis Ii et al., 2016*).

Get_homologues v. 20170418 with MCL clustering was used to cluster genes from 206 *Rhodococcus* genomes into orthologous groups (*Contreras-Moreira and Vinuesa, 2013*). The parse_pangenome_matrix.pl script of get_homologues was used to identify genes enriched (with a 95% threshold) in genomes of *Rhodococcus* in the four plant-associated clades.

To identify pathogenicity loci in sequenced isolates, *fasR*, *fas* and *att* were used as queries in TBLASTN searches against genome assemblies. CONTIGuator was used to map assembled contigs to the reference strain D188 to search for sequences corresponding to pFiD188 or pFID188-like

plasmids (*Galardini et al., 2011*). Get_homologues v. 20170418 was used to cluster genes from each of the virulence plasmids as well as the other *Rhodococcus* linear plasmids (*Francis et al., 2012*). Plasmids were clustered on the basis of gene presence/absence using binary distances and Ward's method for clustering (ward.D2). dbCAN HMMs 5.0 was downloaded and used with *ad hoc* scripts to identify CAZYmes from translated genome sequences (*Yin et al., 2012*). The antiSMASH database was downloaded on 06/2017 and analyzed using antiSMASH ver. 4.0 (*Blin et al., 2017*). Queries used in TBLASTN searches were gene sequences ascertained from searching the literature (*Bruto et al., 2014*; *Glick, 2012*; *Sparacino-Watkins et al., 2014*).

HISAT2 was used to align sequencing reads from strain D188-5 to the D188 reference genome sequence. Variants (SNPs) were called using freebayes, filtered to those with quality score greater than 20 using vcffilter, and annotated using SNPdat v. 1.0.5 (*Doran and Creevey, 2013*).

## Plant growth conditions, assays, and data analysis

Seedling root inhibition assays were performed as described previously, with the exception that after bacteria were adjusted to $OD_{600}$ = 0.5, they were sometimes diluted or concentrated (*Creason et al., 2014b*). At least 100 seedlings were assayed per treatment. Images were taken at 7 days post inoculation (dpi) and data were analyzed. For root hair quantification, a dissecting microscope, equipped with a camera, was used to capture images at 10 and 25 dpi. Root hairs within a 1 cm segment, 1 cm below the stem were quantified using ImageJ (*Schneider et al., 2012*). For cytokinin inhibition assays, three-day-old germinated seedlings were transplanted to MS (half-strength MS, 0.5M MES) medium containing DMSO (control) or 0.01–10.0 μM 6-benzylaminopurine (BA) and then grown and quantified in the same manner.

Leafy galls were induced using the decapitation method on four-week-old *N. benthamiana* plants (*Creason et al., 2014b*). Images were taken 28 dpi.

*Pisum sativum* 'Alaska' seeds were surface-sterilized in 70% ethanol for 1 min, 10% bleach for 10 min, and washed three times with sterile water. Seeds were soaked in sterile water for 60 min and then plated on water agar (15 g agar/L). Plates were incubated at 23°C until radicles were approximately 5 mm. Germinated seeds were soaked in suspension of *Rhodococcus* isolates ($OD_{600}$ = 0.2) or 10 mM $MgCl_2$ buffer for 45 min. Ten seeds per treatment were included in each experiment. Inoculated seeds were placed in sterile test tubes containing 5 ml of Hoagland's nutrient agar. Samples were incubated for 14 days at 23°C with a 16/8 light/dark cycle. Stem number and length were quantified at 14 dpi.

Infection of UCB-1 pistachio was done, with minor modifications, according to previously described protocols (*Stamler et al., 2015b*). Briefly, control isolates, PBTS1, PBTS2, and a 1:1 mixture of PBTS1 and PBTS2 were suspended in 10 mM $MgCl_2$ (final $OD_{600}$ = 0.7). Treatment groups consisting of 15 seedlings were spray-inoculated with 200 ml of bacterial suspension. A mock-inoculated control group was sprayed with 200 ml of 10 mM $MgCl_2$. Inoculated seedlings were placed in humidity chambers for 14 days and the plants were maintained in a greenhouse for seven months. Tree height and internode length were measured at 30 day intervals, and the final measurements were recorded at 210 dpi.

Unless indicated, all experiments were repeated at least three times with similar results. For all data sets, outliers were identified using the ROUT method (Q = 1%) and removed. Data were analyzed using One-way or Two-way ANOVA followed by Tukey's multiple comparisons test (GraphPad Prism v.7, GraphPad Software, La Jolla, CA, USA). Box and whisker plots were generated using the Tukey method; colored dots indicate outliers. Means are indicated by +.

## Nucleic acid manipulations

The *attR* gene was PCR-amplified from D188 genomic DNA and subcloned downstream to the L5 bacteriophage promoter in vector pJDC165 (Jeff Cirillo, Texas A and M). The L5::*attR* construct was verified via Sanger sequencing. *Rhodococcus* competent cells were prepared from overnight-grown 3 ml cultures. Cells were pelleted and washed twice with sterile, cold $dH_2O$, followed by one wash with sterile, cold 10% glycerol. Cells were resuspended in 50 μl 10% glycerol. Plasmid DNA (0.5–1 μg) was added to the cells. After 30 min of incubation on ice, the cells were electroporated in 1 mm gap cuvettes at 2.2 kV. Cells were resuspended in 250 μl SOC medium and incubated at 28°C with shaking for 16 hr prior to plating on LB medium with appropriate antibiotics.

For PCR, the following were used: 1x ThermoPol reaction buffer (New England Biolab, Ipswich, MA, USA), 200 µM dNTPs, 0.2 µM of each primer, 50 ng genomic DNA template, 0.625 units *Taq* DNA polymerase (New England Biolab, Ipswich, MA, USA), in a final volume of 25 µl. PCR conditions were 95°C, 3 min; 30 cycles of 95°C for 30 s, 55°C for 30 s, 72°C for 1 min; 72°C for 10 min; 16°C hold. Reactions with water, instead of a DNA template, were used as a negative control.

For LAMP, the reaction mixture was as follows: 0.5 ng DNA template, 1x ThermoPol reaction buffer (New England Biolab, Ipswich, MA, USA), 5 mM $MgSO_4$, 140 µM dNTPs, 146 µM hydoxynaphthol blue (HNB), 1.6 µM each 16FIP and 16BIP primers, 0.2 µM each 16F3 and 16B3 primers, and 12 U Bst polymerase in a final volume of 25 µl. Reactions were incubated at 64°C for 60 min and then cooled to 4°C. Tubes were centrifuged briefly at 8000 rpm.

RPA was done per the manufacturer's instructions (TwistAmp Basic, TwistDx Limited, Cambridge, UK). Reactions consist of 0.48 µM per primer, 29.5 µl rehydration buffer, 12.2 µl water, and 1.0 µl genomic DNA. A volume of 2.5 µl 280 mM magnesium acetate (MgAc) was added to initiate the reaction. The reaction was incubated at 37°C for 30 min (*Fuller et al., 2017*). Products were purified using the QIAquick PCR purification kit (Qiagen, Germany), run out on a 2.0% agarose gel, stained with ethidium bromide, and visualized under UV light. Products were verified via Sanger sequencing.

RPA reactions coupled to lateral flow detection were comprised of 0.42 µM forward primer, 0.42 µM biotin-labeled reverse primer, 0.12 µM probe, 29.5 µM rehydration buffer, 12.2 µl water, and 1.0 µl of 25 ng/µl genomic DNA. Reactions were added to a freeze-dried pellet provided by the manufacturer (TwistAmp nfo, TwistDx Limited, Cambridge, UK) with the subsequent addition of 2.5 µl of 280 mM MgAc to initiate the reaction. Following subsequent incubation at 37°C for 30 min, the dual-labelled amplicon was visualized using a lateral flow dipstick (Milenia Biotec GMBH, Germany). One microliter of the RPA product was diluted in 49 µl 1.0x PBST and 10 µl of the dilution were applied to the base of the dipstick, which was subsequently submerged in 100 µl 1.0 PBST at room temperature until the visualization of the positive control band, typically lasting two minutes.

Sequences of primers and probes, and their modifications, are described in *Supplementary file 1I*.

## Acknowledgements

This work would not have been possible without the contribution of plants from multiple nurseries, for which we are appreciative. We thank Dr. Edward Davis II for assistance with computational methods, Heidi Lederhos for her assistance in the laboratory, Dr. Zhian Kamvar for assistance with using poppr, Dr. Jeffrey Anderson (Oregon State University) for permitting use of the plate reader, Dr. Jeffery Dangl (University of North Carolina) for providing *Rhodococcus* spp. isolates 114MFTsu3.1, UNC23MFCrub1.1 and 29MFTsu3.1, Dr. Jeff Cirillo (Texas A & M) for pJDC165, Dr. Jennifer Randall (New Mexico State University) for *Rhodococcus* spp. isolates PBTS1, PBTS2, SR18, AGD2M, AGD3B, AGD6D, and AGD6H, Dr. Olivier Vandeputte for D188 carrying pFiD188Δ*att*, and staff in the Center for Genome Research and Biocomputing at Oregon State University, for sequencing services. This work was supported by the National Institute of Food and Agriculture, US Department of Agriculture award 2014-51181-22384, to JHC, MLP, and NJG. EAS was supported by USDA NIFA award 2013-67012-21139. AJW was supported by USDA NIFA award 2017-67012-26126. Last, we thank the Department of Botany and Plant Pathology at Oregon State University for its generous support of SLF and the computing cluster. The funders had no role in study design, data collection and analysis, decision to publish, or preparation of the manuscript.

## Additional information

### Funding

| Funder | Grant reference number | Author |
|---|---|---|
| National Institute of Food and Agriculture | 2014-51181-22384 | Niklaus J Grünwald<br>Melodie L Putnam<br>Jeff H Chang |
| National Institute of Food and Agriculture | 2013-67012-21139 | Elizabeth A Savory |

| National Institute of Food and Agriculture | 2017-67012-26126 | Alexandra J Weisberg |

The funders had no role in study design, data collection and analysis, decision to publish, or preparation of the manuscript.

## Author contributions

Elizabeth A Savory, Conceptualization, Data curation, Software, Formal analysis, Supervision, Funding acquisition, Validation, Investigation, Visualization, Methodology, Writing—original draft, Writing—review and editing; Skylar L Fuller, Conceptualization, Validation, Investigation, Visualization, Methodology, Writing—original draft; Alexandra J Weisberg, Conceptualization, Data curation, Software, Formal analysis, Funding acquisition, Validation, Investigation, Visualization, Methodology, Writing—original draft; William J Thomas, Michael I Gordon, Danielle M Stevens, Validation, Investigation; Allison L Creason, Investigation, Methodology; Michael S Belcher, Investigation; Maryna Serdani, Resources, Data curation, Investigation; Michele S Wiseman, Validation; Niklaus J Grünwald, Conceptualization, Supervision, Funding acquisition, Methodology, Writing—review and editing; Melodie L Putnam, Conceptualization, Resources, Data curation, Supervision, Funding acquisition, Writing—review and editing; Jeff H Chang, Conceptualization, Supervision, Funding acquisition, Visualization, Methodology, Writing—original draft, Project administration, Writing—review and editing

## Author ORCIDs

Alexandra J Weisberg ⓘ http://orcid.org/0000-0002-0045-1368
Michael I Gordon ⓘ http://orcid.org/0000-0002-0176-5924
Niklaus J Grünwald ⓘ https://orcid.org/0000-0003-1656-7602
Jeff H Chang ⓘ http://orcid.org/0000-0002-1833-0695

## Decision letter and Author response

Decision letter https://doi.org/10.7554/eLife.30925.040
Author response https://doi.org/10.7554/eLife.30925.041

# Additional files

## Supplementary files

• Supplementary file 1. Metadata and other supporting information. (A)Characteristics of *Rhodococcus* isolates sequenced in this study. (B) Average nucleotide identity of *Rhodococcus*. (C) Single nucleotide polymorphisms (SNPs) for Clade I isolates of *Rhodococcus*. (D) Single nucleotide polymorphisms (SNPs) for Clade II isolates of *Rhodococcus*. (E) List of 123 genes present in 95% or more plasmid sequences. (F) Characteristics of *Rhodococcus* isolates used in this study. (G) Genes enriched in Clade I–IV members of *Rhodococcus*. (H) Polymorphisms between *Rhodococcus* D188 and D188-5. (I) Sequences of primers and probes used in this study.
DOI: https://doi.org/10.7554/eLife.30925.033
• Transparent reporting form
DOI: https://doi.org/10.7554/eLife.30925.034

## Major datasets

The following dataset was generated:

| Author(s) | Year | Dataset title | Dataset URL | Database, license, and accessibility information |
|---|---|---|---|---|
| Elizabeth A Savory, Skylar L Fuller, Alexandra J Weisberg, William J Thomas, Michael I Gordon, Danielle M Stevens, Allison L Creason, Maryna Serdani, Michele S Wiseman, Niklaus J Grünwald, Melodie L Putnam, Jeff H Chang | 2017 | Sequencing of plant-associated Rhodococcus | https://www.ncbi.nlm.nih.gov/bioproject/PRJNA395383/ | Publicly available at NCBI BioProject (accession no: PRJNA395383) |

The following previously published dataset was used:

| Author(s) | Year | Dataset title | Dataset URL | Database, license, and accessibility information |
|---|---|---|---|---|
| A L Creason, O M Vandeputte, E A Savory, E W Davis II, et al | 2014 | Analysis of Genome Sequences from Plant Pathogenic Rhodococcus Reveals Genetic Novelties in Virulence Loci | https://www.ncbi.nlm.nih.gov/genome/?term=PRJNA233522 | Publicly available at NCBI Genome (accession no: PRJNA233522) |

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
