## [Decision Letter]

Thank you for submitting your article "Evolutionary transitions between beneficial and phytopathogenic *Rhodococcus* challenge disease management" for consideration by *eLife*. Your article has been reviewed by three peer reviewers, and the evaluation has been overseen by a Reviewing Editor and Christian Hardtke as the Senior Editor. The following individuals involved in review of your submission have agreed to reveal their identity: Cara H Haney (Reviewer #1); Boris Vinatzer (Reviewer #2).

The reviewers have discussed the reviews with one another and the Reviewing Editor has drafted this decision to help you prepare a revised submission.

Summary:

The work explored the evolution of virulence in Rhodococcus sp. through an ecological survey, they sequence genomes and look for bacterial loci that correlate with symbiosis versus pathogenesis. This work provides molecular evidence for the evolutionary transition from symbiont to pathogen, and shows that a single plasmid/gene cluster is necessary and sufficient to turn a symbiont into a pathogen. This work has implications for evolutionary biology, commensalism versus pathogenesis and is directly useful for agriculture. The study is well done and well controlled and the data are in general quite convincing. All the reviewers were quite positive about this study. They agree that most of the conclusions are justified.

Essential revisions:

Several issues came up that need to be addressed.

1) Data analysis and presentation should be improved:

A) It’s not clear whether virulence genes are always carried on a plasmid in Rhodococcus. Most of the genomes present in this study are in 10-100 contigs, thus the bulk of the genomic data is in contigs, not circularized chromosomes or plasmids. Based on the data presented alone, it is possible that the virulence genes are present in genomic island configurations in some strains rather than on plasmids. While D188 clearly has a virulence plasmid mobilizable to Clade 1 organisms, this may not be the case for all of the draft genomes.

This could be resolved either by the authors showing that the sequences containing virulence genes can always be assembled into circular plasmids, providing evidence the virulence genes are higher copy (which would suggest they're on plasmids) OR by rewriting the text and remaking the figure legends to indicate that the focus is on presence/absence of virulence genes rather than the presence/absence of a plasmid.

B) Figure 2 is difficult to interpret. The authors should reconsider how they present these data. One possibility is a scatterplot showing pairwise SNP divergences for the chromosome on one axis and some metric for virulence plasmid relatedness on the other (e.g. correlation coefficient of gene presence/absence). Then, symbols and colors could be used to add additional information about the spatial and temporal "patterns" of interest.

C) Some details in the graphs were not warranted (and indeed not used by the authors as well). For example, the authors only commented on the relationship between Chromosome (Clade ID) and Plasmid (Type ID) without referring to fine details in Figure 3. It was difficult to comprehend the figures with so many broken lines crossing each other. Why not just present a cross-table [two factors] to present number of strains in each category – this should be able to clearly show non-congruence?

D) The authors presented the statistical results early in figures. Please also include the p-value in the text as well.

2) The writing and data presentation needs to be improved.

The introduction should be more specific about the research objectives. These were only obvious upon reading the Discussion section when the authors point-by-point demonstrated misdiagnosis of the disease causal agent in previous work. Please work on the Results and Discussion sections about "patterns" in the population structure. Please tone down the conclusions in regard to the pistachio decline. The authors provide convincing information that the two isolates were not pathogenic and that this is most likely not due to a loss of the virulence plasmid during growth in the laboratory after the original isolation. However, the authors make a point how pathogenic and non-pathogenic strain of Rhodococcus can co-exist on plants. Therefore, isn't it possible that other pathogenic Rhodococcus strains were present on pistachio plants and did in fact cause the decline? Although this is unlikely based on all the circumstantial evidence that argues against Rhodococcus as pathogenic agent of the pistachio decline, but the authors could still mention this as a possibility.

3) Claims of benefit:

While plant growth promotion is often beneficial, the authors should limit claims of benefit to the discussion and limit the description in the results to be more objective. It seems possible the observed morphological changes in response to non-pathogenic Rhodococcus are not in fact beneficial, but rather an intermediate pathogenic phenotype. The authors do not show benefit directly, but rather morphological changes to root hair density and length, which could be beneficial under certain circumstances.

4) Suggestion for removing some data:

Subsection “Isolates of *Rhodococcus* lacking the functional set of virulence genes are beneficial bacteria” – We suggest removing this section and corresponding data from the manuscript. There isn't clear evidence (that the reviewers knew of) where callose deposition by heat killed bacteria has been shown to be a good readout or proxy for microbial effects on systemic plant immunity, or priming. The authors definitely cannot conclude from this result that microbial antagonism is a more likely to be the cause of biocontrol. It could be that live bacteria are required for modulation of systemic immunity, or that the Rhodococcus suppresses local defense responses but induces systemic defenses. (Indeed, the bacterial ISR strain *Pseudomonas* simiae WCS417 induces systemic defenses but suppresses local defenses; see Millet et al., 2010).

---

## [Author Response]

Essential revisions:Several issues came up that need to be addressed.1) Data analysis and presentation should be improved:A) It’s not clear whether virulence genes are always carried on a plasmid in Rhodococcus. Most of the genomes present in this study are in 10-100 contigs, thus the bulk of the genomic data is in contigs, not circularized chromosomes or plasmids. Based on the data presented alone, it is possible that the virulence genes are present in genomic island configurations in some strains rather than on plasmids. While D188 clearly has a virulence plasmid mobilizable to Clade 1 organisms, this may not be the case for all of the draft genomes.

This is an excellent point. In 62 of the 66 assemblies (virulence gene-carrying isolates; past and current efforts), the virulence genes are present on a single large contig that have strong evidence supporting it corresponds to the linear virulence plasmid. These contigs have plasmid features (pFi_009 was used as a representative marker gene; telomere-associated protein) and are colinear to the reference virulence plasmid of D188. Two assemblies had the virulence loci and pFi_009 on separate contigs, but the contigs are similar in composition to the reference plasmid sequence, are co-linear to the reference, and carry features unique to plasmids. These 64 are distinctly different from A21d2 and A25f, which appears to have virulence loci in the chromosome. BLAST using virulence genes as queries only identified contigs that corresponded to the linear virulence plasmid.

This could be resolved either by the authors showing that the sequences containing virulence genes can always be assembled into circular plasmids, providing evidence the virulence genes are higher copy (which would suggest they're on plasmids) OR by rewriting the text and remaking the figure legends to indicate that the focus is on presence/absence of virulence genes rather than the presence/absence of a plasmid.

The average relative coverage (linear plasmid:chromosome) of sequencing was 1.89 ± 0.56. Only three assemblies had a relative coverage < 1.0 but each of these three had the virulence genes and pFi_009 on the same contig. The virulence plasmid is not a typical plasmid. It is ~200 kb in length and linear. It is clear we did a poor job communicating these details so we added a sentence to the introduction that conveys this information.

We hope the newly added data communicate the confidence in our conclusions regarding the virulence plasmids. Because of the mobility of the plasmids, their distribution and presence/absence are important to the narrative of this manuscript.

B) Figure 2 is difficult to interpret. The authors should reconsider how they present these data. One possibility is a scatterplot showing pairwise SNP divergences for the chromosome on one axis and some metric for virulence plasmid relatedness on the other (e.g. correlation coefficient of gene presence/absence). Then, symbols and colors could be used to add additional information about the spatial and temporal "patterns" of interest.

We kept the minimum spanning network because this is a standard visual for genomic epidemiology. We suspect the isolate and plasmid data made the figure unnecessarily complex. We removed strain, date, and plasmid information. We constructed panels C-G to focus on only the SNP genotypes of interest. Plasmid information was simplified by using lines to represent the different plasmid types and variants.

C) Some details in the graphs were not warranted (and indeed not used by the authors as well). For example, the authors only commented on the relationship between Chromosome (Clade ID) and Plasmid (Type ID) without referring to fine details in Figure 3. It was difficult to comprehend the figures with so many broken lines crossing each other. Why not just present a cross-table [two factors] to present number of strains in each category – this should be able to clearly show non-congruence?

Based on the description of the data, we believe the comment pertains to Figure 3—figure supplement 2. The purpose of Figure 3—figure supplement 2 is to show that plasmid types are not restricted to clades of *Rhodococcus*. We added new text and highlighted some examples to support our conclusion that the plasmid is horizontally acquired and shared between the different clades of *Rhodococcus*. With regard to finer details, those relevant to the MLSA tree were described when Figure 1 was presented. Because of polytomies in the tree of the plasmid genes, we could not elaborate on fine details.

D) The authors presented the statistical results early in figures. Please also include the p-value in the text as well.

Addressed.

2) The writing and data presentation needs to be improved.The introduction should be more specific about the research objectives. These were only obvious upon reading the Discussion section when the authors point-by-point demonstrated misdiagnosis of the disease causal agent in previous work. Please work on the Results and Discussion about "patterns" in the population structure. Please tone down the conclusions in regard to the pistachio decline. The authors provide convincing information that the two isolates were not pathogenic and that this is most likely not due to a loss of the virulence plasmid during growth in the laboratory after the original isolation. However, the authors make a point how pathogenic and non-pathogenic strain of Rhodococcus can co-exist on plants. Therefore, isn't it possible that other pathogenic Rhodococcus strains were present on pistachio plants and did in fact cause the decline? Although this is unlikely based on all the circumstantial evidence that argues against Rhodococcus as pathogenic agent of the pistachio decline, but the authors could still mention this as a possibility.

We changed the word “pattern” to “links” and changed the relevant text.

The reviewers are correct. Previously in the introduction, we had a sentence that informed readers that we would examine the previous experiments that led to the conclusion that *Rhodococcus* is the causative agent of pistachio bushy top syndrome. We removed that sentence and replaced it with a phrase that clearly states one objective of this manuscript is to reexamine previous conclusions.

We understand the need to exercise caution and toned down the language regarding our claim of a misdiagnosis. It may be helpful to describe the depth to which we have investigated this problem and the process we followed. We feel it is important to show that the reported data represent a fraction of our efforts, and to impress upon the reviewers that we understand the gravity of the situation, and prior to submitting our work, attempted to preemptively limit potential harm to all involved parties. We are concerned with the effects of the results from previous studies on US agriculture and the potential effects on public trust in science. We are also concerned with the potential repercussions of our study on colleagues.

We agree with the reviewers that the hypothesis that *Rhodococcus* is not a pathogen of pistachio is untestable. There is a possibility that pathogenic *Rhodococcus* can cause disease to pistachio. But it would more likely be leafy galls or witches’ brooms, as has been documented for all previous woody hosts infected with pathogenic *Rhodococcus*, and opposed to the phenotypically distinct “pistachio bushy top syndrome”. To test previous conclusions, we can only uncover and support possible alternative explanations (Figure 4–Figure 7), challenge the reproducibility of results (Figure 7–Figure 8; discussion), and question experimental design (Figure 7; Discussion section). Based on results from the approaches that are available to us, we concluded “pistachio bushy top syndrome” was misdiagnosed.

We were first made aware of the problem in April of 2014. Multiple attempts to reproduce the detection of virulence genes and demonstrate causation of disease were done independently by different researchers in both the Putnam and Chang groups. From June 2014 to December 2015, we had nine email communications to inform the corresponding author of Stamler et al., (2015) that we were unable to reproduce the findings that: (1) virulence genes are encoded by pistachio-associated *Rhodococcus*, (2) pathogenic (those that encode virulence genes) *Rhodococcus* could be cultured from pistachio (110 plants were assayed, including those in tissue culture, from nurseries, and from fields), or (3) the pistachio-associated isolates cultured from pistachio cause disease to plants. We also cautioned against the use of *vicA* for molecular detection of pathogenic isolates because it does not discriminate pathogenic from non-pathogenic isolates. The email communications included official reports, prepared by the Oregon State University Plant Clinic, directed by a co-author of our study. In total, at least 22 people, including authors of the original publications, extension agents, stakeholders, and members of the California Pistachio Research Board, have been informed of our results. Most of the data were shared in 2014, prior to publication of Stamler et al., (2015), and some are in this submitted manuscript.

Reproducibility is one of the central tenants of the scientific method. In most instances, the research community has the luxury of moving past irreproducible results, and the publications reporting such results receive little attention. However, the diagnosis of “pistachio bushy top syndrome” is not academic. The results that members of *Rhodococcus* cause “pistachio bushy top syndrome”, which we were unable to reproduce, are being used to guide the action of stakeholders, actions that have not only severe economic consequences, but also legal consequences. This compelled us to be firmer in our conclusions.

3) Claims of benefit:While plant growth promotion is often beneficial, the authors should limit claims of benefit to the discussion and limit the description in the results to be more objective. It seems possible the observed morphological changes in response to non-pathogenic Rhodococcus are not in fact beneficial, but rather an intermediate pathogenic phenotype. The authors do not show benefit directly, but rather morphological changes to root hair density and length, which could be beneficial under certain circumstances.

We agree and exercised more caution in the Results section and tempered our conclusions in the Discussion section.

4) Suggestion for removing some data:Subsection “Isolates of Rhodococcus lacking the functional set of virulence genes are beneficial bacteria” – We suggest removing this section and corresponding data from the manuscript. There isn't clear evidence (that the reviewers knew of) where callose deposition by heat killed bacteria has been shown to be a good readout or proxy for microbial effects on systemic plant immunity, or priming. The authors definitely cannot conclude from this result that microbial antagonism is a more likely to be the cause of biocontrol. It could be that live bacteria are required for modulation of systemic immunity, or that the Rhodococcus suppresses local defense responses but induces systemic defenses. (Indeed, the bacterial ISR strain Pseudomonas simiae WCS417 induces systemic defenses but suppresses local defenses; see Millet et al., 2010).

These data were removed.